# Small molecule proteostasis regulators that reprogram the ER to reduce extracellular protein aggregation

Lars Plate[1,2†], Christina B Cooley[1,2†‡], John J Chen[2], Ryan J Paxman[1], Ciara M Gallagher[3], Franck Madoux[4,5§], Joseph C Genereux[1,2¶], Wesley Dobbs[6], Dan Garza[6], Timothy P Spicer[4,5], Louis Scampavia[4,5], Steven J Brown[7], Hugh Rosen[2,7,8], Evan T Powers[1], Peter Walter[3], Peter Hodder[4,5§], R Luke Wiseman[2,8*], Jeffery W Kelly[1,2*]

[1]Department of Chemistry, The Scripps Research Institute, La Jolla, United States; [2]Department of Molecular and Experimental Medicine, The Scripps Research Institute, La Jolla, United States; [3]Department of Biochemistry and Biophysics, Howard Hughes Medical Institute, University of California, San Francisco, United States; [4]The Scripps Research Institute Molecular Screening Center, Translational Research Institute, Jupiter, United States; [5]Lead Identification Division, Translational Research Institute, Jupiter, United States; [6]Proteostasis Therapeutics Inc, Cambridge, United States; [7]The Scripps Research Institute Molecular Screening Center, La Jolla, United States; [8]Department of Chemical Physiology, The Scripps Research Institute, La Jolla, United States

*For correspondence: wiseman@scripps.edu (RLW); jkelly@scripps.edu (JWK)

†These authors contributed equally to this work

Present address: ‡Trinity University, San Antonio, United States; §Amgen, Inc, Thousand Oaks, United States; ¶University of California, Riverside, Riverside, United States

**Abstract** Imbalances in endoplasmic reticulum (ER) proteostasis are associated with etiologically-diverse degenerative diseases linked to excessive extracellular protein misfolding and aggregation. Reprogramming of the ER proteostasis environment through genetic activation of the Unfolded Protein Response (UPR)-associated transcription factor ATF6 attenuates secretion and extracellular aggregation of amyloidogenic proteins. Here, we employed a screening approach that included complementary arm-specific UPR reporters and medium-throughput transcriptional profiling to identify non-toxic small molecules that phenocopy the ATF6-mediated reprogramming of the ER proteostasis environment. The ER reprogramming afforded by our molecules requires activation of endogenous ATF6 and occurs independent of global ER stress. Furthermore, our molecules phenocopy the ability of genetic ATF6 activation to selectively reduce secretion and extracellular aggregation of amyloidogenic proteins. These results show that small molecule-dependent ER reprogramming, achieved through preferential activation of the ATF6 transcriptional program, is a promising strategy to ameliorate imbalances in ER function associated with degenerative protein aggregation diseases.

## Introduction

Nearly 1/3 of the human proteome is targeted to the endoplasmic reticulum (ER) for folding and trafficking to downstream secretory pathway environments, including the extracellular space. These secreted proteins engage ER-localized protein homeostasis (or proteostasis) factors such as chaperones and folding enzymes to facilitate their folding (*Brodsky and Skach, 2011*; *Powers et al., 2009*). Proteins unable to fold in the ER are identified by ER proteostasis factors that direct them towards the proteasome (via ER-associated degradation) or lysosome for degradation (*Brodsky and*

**eLife digest** Newly made proteins must be folded into specific three-dimensional shapes before they can perform their roles in cells. Many proteins are folded in a compartment called the endoplasmic reticulum before being transported to their final location. However, if a cell suddenly needs to make a large number of new proteins, it can overwhelm the endoplasmic reticulum and unfolded proteins may accumulate. The cell responds to this stress by activating the unfolded protein response, which increases the folding capacity of the endoplasmic reticulum to match the demand. However, if the stress persists, then the unfolded protein response instructs the cell to die to protect the rest of the body.

A protein called ATF6 is involved in one branch of the unfolded protein response. Endoplasmic reticulum stress causes ATF6 to move from the endoplasmic reticulum to another cell compartment where certain enzymes are able to cut the protein. A fragment of ATF6 then moves to the nucleus of the cell to activate genes needed to increase the cell's capacity to fold proteins.

Errors in protein folding can cause serious diseases in humans and other animals. Drugs that target ATF6 might be able to regulate part of the unfolded protein response to treat these diseases. However, no drugs that act on ATF6 had been identified. Now, two groups of researchers have independently identified small molecules that specifically target ATF6.

Plate et al. used a new approach to screen over 600,000 small molecules and identified a small number that could activate ATF6-regulated genes without inducing global endoplasmic reticulum stress. Further experiments tested whether any of these ATF6 drug candidates could prevent the release of incorrectly folded versions of two particular proteins from cells that are associated with types of amyloid disease in humans. One of the small molecules tested effectively reduced the release of these proteins and prevented harmful deposits of the proteins forming in the spaces surrounding cells.

In an independent study, Gallagher et al. identified a type of small molecule that can inhibit the activity of ATF6. Together, these findings may lead to further development of new drugs for treating diseases associated with incorrect protein folding in the endoplasmic reticulum.

*Skach, 2011*; *Smith et al., 2011*). This partitioning toward degradation pathways prevents the folding and trafficking of destabilized, misfolding-prone proteins from the ER, which could aggregate into proteotoxic conformations in downstream secretory environments (*Brodsky and Skach, 2011*; *Chen et al., 2011*; *Powers et al., 2009*).

Many degenerative diseases result from imbalances in ER proteostasis (*Hipp et al., 2014*; *Luheshi et al., 2008*). The aberrant secretion of properly folded, but destabilized aggregation-prone proteins facilitates extracellular proteotoxic aggregation associated with many amyloid diseases (*Tipping et al., 2015*). Examples include the transthyretin (TTR) or light chain (AL) amyloidoses (*Blancas-Mejia and Ramirez-Alvarado, 2013*). Similarly, the inability to efficiently degrade destabilized, aggregation-prone variants of secreted proteins such as rhodopsin and α-1-antitrypsin (A1AT) facilitates their proteotoxic intracellular aggregation linked to retinitis pigmentosa and liver disease, respectively (*Gooptu et al., 2014*; *Lin and Lavail, 2010*).

One strategy to ameliorate the above-mentioned and related diseases is to transcriptionally reprogram the ER proteostasis network (ER PN), which has the potential to increase the ER's capacity to identify destabilized, misfolding-prone proteins and prevent their secretion to downstream secretory environments (*Calamini and Morimoto, 2012*; *Chen et al., 2015*; *Gestwicki and Garza, 2012*). ER PN remodeling can be achieved by activation of the unfolded protein response (UPR), comprising three signaling pathways activated downstream of the ER stress sensing proteins IRE1, ATF6, and PERK (*Walter and Ron, 2011*). In response to the accumulation of misfolded proteins within the ER (i.e., ER stress), these sensors are activated, resulting in ER PN reprogramming predominantly mediated by the UPR-associated transcription factors XBP1s and ATF6. These transcription factors induce distinct, but overlapping, transcriptional programs that alter the composition of ER PN pathways (*Adachi et al., 2008*; *Shoulders et al., 2013*; *Yamamoto et al., 2004*). Intriguingly, preferential ATF6-mediated reprogramming of the ER PN offers a unique opportunity to alter the

fate of disease-associated proteins. Genetic activation of the ATF6 transcriptional program attenuates intracellular accumulation of mutant rhodopsin and the Z-variant of A1AT (*Chiang et al., 2012*; *Smith et al., 2011*). Furthermore, stress-independent chemical genetic activation of a ligand-regulated ATF6 attenuates the secretion and extracellular aggregation of destabilized, amyloidogenic variants of TTR or immunoglobulin light chains (LCs), without significantly impacting secretion of stable, non-amyloidogenic variants of these proteins or the global endogenous secreted proteome (*Chen et al., 2014*; *Cooley et al., 2014*; *Shoulders et al., 2013*). These results suggest that small molecules that similarly induce preferential ER PN remodeling through pharmacologic activation of the ATF6 transcriptional program should attenuate the aberrant secretion and aggregation of these proteins linked to degenerative diseases (*Chen et al., 2015*).

Traditionally, UPR signaling is activated using small molecule ER stressors such as the SERCA inhibitor thapsigargin (Tg) or tunicamycin (Tm), an inhibitor of N-linked glycosylation (*Chen et al., 2015*). However, chronic addition of these ER stressors induces apoptosis, predominantly through activation of the PERK arm of the UPR (*Hetz, 2012*; *Tabas and Ron, 2011*). Small molecule kinase inhibitors and ATP mimetics have been shown to activate the IRE1/XBP1s arm of the UPR (*Mendez et al., 2015*; *Papa et al., 2003*; *Wang et al., 2012*). The small molecule BIX induces expression of the ATF6-target gene *BiP* through an ATF6-dependent mechanism, but does not significantly induce expression of other ATF6 target genes such as *GRP94, p58^{IPK}* and *PDIA4*, likely limiting its utility for ER PN reprogramming in the above-mentioned diseases (*Kudo et al., 2008*). These results highlight the need for further development of molecules that preferentially activate UPR-associated transcriptional programs, such as that regulated by ATF6, to define the benefit of ER PN reprogramming to ameliorate protein aggregation diseases and protein misfolding diseases.

A challenge in identifying such compounds is the difficulty associated with efficiently assessing the preferential activation of a sub-transcriptional pathway that is part of a highly integrated signaling network (e.g., the ATF6 arm of the UPR). An increasing proportion of failed clinical trials have highlighted the importance of the early elimination of compounds functioning through off-target effects, which can be achieved using multidimensional assays such as gene expression profiling (*Feng et al., 2009*; *Verbist et al., 2015*). The use of a transcriptional profiling approach early in the drug discovery process and before functional assessment allows for the identification of promising small molecules that preferentially activate a target transcriptional program, instead of global network activation. However this strategy has only been employed in a small number of drug development programs and has not been used to identify small molecule activators of a sub-UPR transcriptional program, such as that regulated by ATF6 (*Antipova et al., 2008*; *Stegmaier et al., 2004*; *Stegmaier et al., 2007*).

Here, we employ a cell-based high-throughput screen (HTS) in combination with transcriptional profiling to identify non-toxic small molecules that preferentially activate the ATF6 transcriptional program, independent of global UPR activation. Small molecule ER proteostasis regulators identified by our screening strategy mimic the ER PN reprogramming afforded by chemical genetic ATF6 activation, through a process dependent on endogenous ATF6. Furthermore, our small molecule ER proteostasis regulators phenocopy the ability of chemical genetic ATF6 activation to preferentially decrease the secretion and proteotoxic aggregation of destabilized, amyloidogenic proteins without affecting the secretion of stable, non-amyloidogenic proteins or the global endogenous secreted proteome. These results highlight the utility of transcriptional profiling upstream of functional studies as part of a multi-tiered screening strategy to identify small molecule ER proteostasis regulators that can be employed to ameliorate imbalances in ER function involved in aggregation-associated degenerative diseases and possibly loss-of-function protein misfolding diseases.

## Results

### Cell-based high-throughput screen to identify small molecule ER proteostasis regulators

We employed a three-tiered screening strategy to identify small molecules that preferentially activate the ATF6 UPR transcriptional program for ER PN reprogramming (*Figure 1A*). First, a cell-based HTS was performed using a transcriptional reporter that includes a fragment of the UPR-inducible *BiP* promoter driving expression of firefly luciferase (ERSE-FLuc; *Figure 1B*) (*Yoshida et al., 1998*).

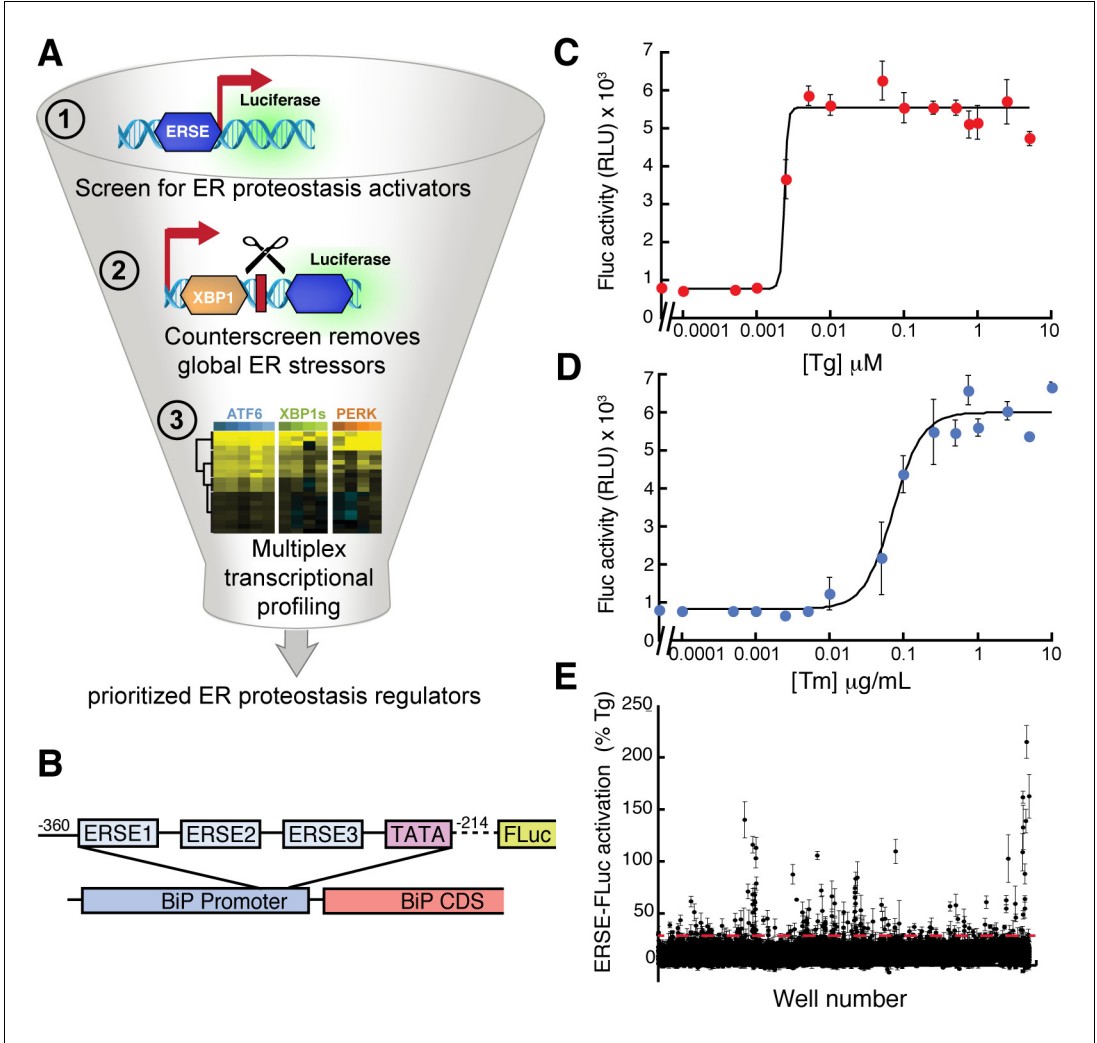

**Figure 1.** High-throughput screen to identify small molecule ER proteostasis regulators. (**A**) Illustration showing the three-tiered screening strategy implemented to identify small molecules that preferentially activate the ATF6 transcriptional program. (**B**) Schematic of the ERSE-firefly luciferase (FLuc) reporter used in our HTS approach. (**C**) Activation of FLuc luminescence in HEK293T-Rex cells stably expressing ERSE-FLuc treated with the indicated concentrations of thapsigargin (Tg) for 18 hr. Error bars represent standard deviation for n = 3 replicates. (**D**) Activation of FLuc luminescence in HEK293T-Rex cells stably expressing ERSE-FLuc treated with the indicated concentrations of tunicamycin (Tm) for 18 hr. Error bars represent standard deviation for n = 3 replicates. (**E**) Plot showing ERSE-FLuc activation in HEK293T-Rex cells stably expressing ERSE-FLuc treated with the 13,748 small molecule ER proteostasis activators identified in the primary screen (6.8 µM; 18 hr). Luminescence is shown as % signal relative to Tg treatment (500 nM; 18 hr). Error bars show standard deviation for n = 3 replicates. The dashed red line shows 25.1% Tg activity.
The following figure supplement is available for figure 1:

**Figure supplement 1.** Selectivity of the ERSE-FLuc reporter for the ATF6 UPR arm and highly represented chemical substructures in the top 281 ER proteostasis regulators.

*BiP* is preferentially induced by ATF6 (*Shoulders et al., 2013*), indicating that the ERSE-FLuc reporter should preferentially report on activation of the ATF6 transcriptional program. We tested the dependence of ERSE-FLuc activation on XBP1s and ATF6 in HEK293[DAX] cells that stably express tet-inducible XBP1s and a trimethoprim (TMP)-regulated dihydrofolate reductase (DHFR)-ATF6 fusion, hereafter referred to as chemical genetic ATF6 activation (*Shoulders et al., 2013*). As predicted, the ERSE-FLuc reporter was preferentially activated by ATF6, relative to XBP1s (*Figure 1—figure supplement 1A*) in HEK293[DAX] cells. We then stably transfected the ERSE-FLuc reporter into HEK293T-Rex cells and selected a single clone exhibiting dose-dependent reporter activation upon

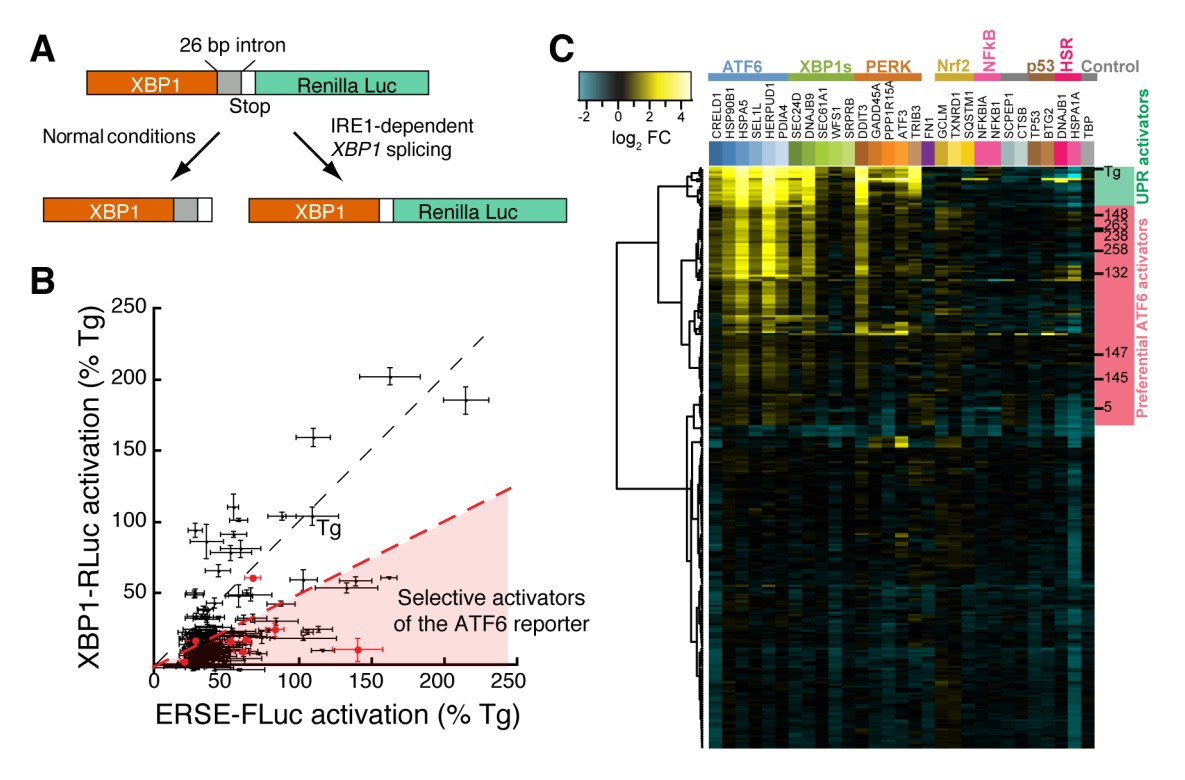

**Figure 2.** Counterscreening and transcriptional profiling identifies preferential activators of the ATF6 transcriptional program. (**A**) Schematic of the XBP1-RLuc splicing reporter used to monitor activation of the IRE1/XBP1s arm of the UPR in HEK293T-Rex cells treated with small molecule ER proteostasis regulators. (**B**) Plot of ERSE-FLuc activation and XBP1-RLuc activation for the top 281 small molecule ER proteostasis regulators (6.8 μM; 18 hr) in HEK293T-Rex cells stably expressing ERSE-FLuc or XBP1-RLuc. Activation for each axis is shown as the percent signal relative to Tg treatment (500 nM; 18 hr). Prioritized ER proteostasis regulators described in *Table 1* are shown in red. Error bars show standard deviation for n = 3. (**C**) Heat map of multiplex gene expression (MGE) profiling for the top 281 ER proteostasis regulator compounds. HEK293T-Rex cells were incubated with 10 μM of each proteostasis regulator for 6 hr, then cells were lysed and MGE profiling was performed. Genes probed constitute transcriptional targets preferentially induced by the ATF6, XBP1s or PERK arms of the UPR or other stress-responsive signaling pathways, as indicated. Heat map is shown with compounds clustered according to their activity. The green section represents global ER stressors while the pink sections represent molecules that preferentially activate the ATF6 transcriptional program. Prioritized ER proteostasis regulators identified in *Table 1* are highlighted.

The following source data and figure supplement are available for figure 2:

**Source data 1.** Excel spreadsheet showing the structure, % activation ERSE-FLuc, and % activation XBP1-RLuc for the top 281 compounds identified through primary HTS screening and confirmation screening.

**Source data 2.** Excel spreadsheet showing the Multiplex Gene Expression (MGE) expression data used to prepare *Figure 2C* and *Figure 2—figure supplement 1C*.

**Figure supplement 1.** XBP1-RLuc reporter dose response and Tg-normalized MGE profiling for 281 top ER proteostasis regulators.

treatment with the ER stressors Tg or Tm (*Figure 1C,D*). This assay was further miniaturized for 1536-well high-throughput screening at the Scripps Research Institute Molecule Screening Center (SRIMSC) (*Supplementary file 1*).

We screened the 644,951-small molecule Scripps Drug Discovery Library (SDDL) at SRIMSC to identify molecules that activate the ERSE-FLuc reporter. The performance of this assay was consistent across all experimental plates (Z' = 0.58 ± 0.05) and exhibited a robust signal to noise ratio (signal/background = 6.21 ± 0.73) (*Supplementary file 1*). Small molecule activation of ERSE-FLuc was normalized to Tg (assigned to be 100% activation), allowing comparisons between screening plates. This screen identified 13,799 molecules that activated the ERSE-FLuc reporter >25.1%. These hits were then filtered against results from a previous screen of the SDDL to remove 49 small molecules

that activate the cytosolic heat shock response (*Calamini et al., 2012*). Confirmation screening of the remaining 13,750 compounds identified 12,376 molecules that activated the ERSE-FLuc reporter 3 standard deviations above the DMSO control (hit cutoff 5.7% activation)–a 90% hit confirmation (*Figure 1E*). To decrease the number of compounds for follow-up, we increased the cutoff stringency to that used in the primary screen (25.1% activation), which narrowed the list of ERSE-FLuc activators to 281 compounds (*Figure 1E*, *Figure 2—source data 1*). These include the ER stressors Tg and Tm, which were both present in the SDDL. All 281 confirmed hits were subjected to quality control at SRIMSC to confirm identity and purity using liquid chromatography/mass spectrometry. A maximum common substructure search identified 12 chemical substructures that were highly represented in these 281 ERSE-FLuc activators (*Figure 1—figure supplement 1B,C*). These include catechols (64/281), anilides (61/281) and benzylidene hydrazines (33/281).

## Counterscreening excludes molecules that induce ER stress and/or globally activate the UPR

We next sought to identify small molecule ER proteostasis regulators that preferentially activate the ATF6 transcriptional program independent of global ER stress. To remove molecules that induce ER stress and global UPR activation, the top 281 compounds were counterscreened using an alternative luciferase reporter, signifying activation of the IRE1/XBP1s arm of the UPR (*Figure 1A*). This reporter contains Renilla luciferase (RLuc) expressed out of frame downstream of the XBP1 splice site, preventing RLuc translation in the absence of ER stress (*Figure 2A*) (*Back et al., 2006*; *Iwawaki et al., 2004*). In response to ER stress-dependent IRE1 activation, the 26-nt *XBP1* intron is removed, producing a frame shift that allows for RLuc translation and luminescence. Robust dose-dependent activation of the XBP1-RLuc reporter upon addition of the ER stressors Tg and Tm was confirmed in HEK293T-Rex cells stably expressing the XBP1-RLuc reporter, producing a Z' score of 0.75 for Tg (*Figure 2—figure supplement 1A,B*).

We compared the activation of this XBP1-RLuc reporter with that observed for the ERSE-FLuc reporter, both being normalized to the Tg control (defined as 100% activation). Importantly, Tg (1 of the 281 screening hits) robustly activates both the ERSE-FLuc and XBP1-Rluc reporters, confirming the ability of this approach to identify global ER stressors (*Figure 2B*). In contrast, the majority of molecules identified in the primary screen (200/281) activate the ERSE-FLuc reporter >2-fold better than the XBP1s-Rluc reporter (*Figure 2B*). This suggests that small molecules identified from the HTS preferentially activate the ATF6 transcriptional program independent of ER stress and/or global UPR activation.

## Transcriptional profiling identifies molecules that preferentially activate the ATF6 program

As a final filter, we performed multiplex gene expression (MGE) profiling in HEK293T-Rex cells treated with the 281 hits to directly monitor transcriptional changes and identify compounds that preferentially active the ATF6 transcriptional program (*Figure 1A*). While the ATF6 and XBP1s luciferase reporters were indispensible for initial high-throughput screening, we wanted to ensure that any further prioritization was based on a comprehensive picture of the transcriptional remodeling induced by the compounds. For instance, we wanted to make certain that the compounds induce multiple ATF6 targets in addition to *BiP*, as well as monitor that no other stress-responsive signaling pathways are affected. For this approach, we defined 5–6 target genes that were previously identified as preferential targets of the ATF6, XBP1s or PERK signaling pathway (*Lu et al., 2004*; *Shoulders et al., 2013*), as well as genes downstream of other stress responsive signaling pathways, including the heat shock response (HSR), the oxidative stress response (Nrf2), and inflammatory signaling (NFκB) (see *Figure 2—source data 2* for a complete list of genes). The MGE analysis quantifies mRNA levels of these target genes in HEK293T-Rex cells treated with our prioritized 281 ER proteostasis regulators. This provides a medium-throughout approach to define transcriptional changes induced by our screening hits.

We performed hierarchical clustering of the 281 screening hits based on the MGE data (*Figure 2C*). This showed that compounds grouped into three distinct transcriptional profiles: 1) global UPR activators that activate ATF6, XBP1s and PERK target genes (green), 2) preferential activators of ATF6 targets (pink), and 3) compounds with weak to no transcriptional effect. Apart from

**Table 1.** Prioritized ER proteostasis regulators.

| Name | Structure | MW | Polar Surface Area | AlogP | Structural moieties (*Figure 1—figure supplement 1B–C*) | % ERSE-FLuc activation* | % XBP1-RLuc activation* | Promiscuity** |
|---|---|---|---|---|---|---|---|---|
| 5 | | 271.2 | 41.13 | 2.62 | - | 60.9 ± 6.0 | 9.1 ± 0.7 | 1 out of 31 |
| 132 | | 283.3 | 58.56 | 3.27 | F | 140.1 ± 17.4 | 10.3 ± 7.9 | 2 out of 32 |
| 145 | | 260.3 | 75.36 | 3.01 | F, H | 83.4 ± 6.3 | 24.5 ± 4.7 | 1 out of 31 |
| 147 | | 255.3 | 49.33 | 3.35 | B, F | 63.2 ± 0.6 | 17.5 ± 1.9 | 1 out of 32 |
| 148 | | 322.3 | 149.12 | 2.00 | B, F | 52.5 ± 14.3 | 16.6 ± 2.4 | 3 out of 31 |
| 238 | | 314.7 | 69.64 | 2.94 | B, L | 68.2 ± 5.7 | 61.1 ± 0.1.5 | 2 out of 31 |
| 258 | | 393.2 | 91.15 | 3.76 | C, D | 28.6 ± 6.1 | 32.6 ± 1.0 | 1 out of 31 |
| 263 | | 257.2 | 90.44 | 3.10 | C | 20.7 ± 2.3 | 2.4 ± 1.0 | 1 out of 31 |

*% Tg activation from triplicate confirmation screen at 6.8 μM

** Number of screening assays in which the compound was found active

the UPR, no compounds significantly induced gene targets regulated by other stress-responsive signaling pathways (*Figure 2C*). In order to compare the relative induction of UPR-associated transcriptional programs activated downstream of ATF6, XBP1s and PERK, we normalized transcript changes to those observed in Tg-treated cells (*Figure 2—figure supplement 1C*). This allows direct comparison of transcriptional reprogramming across the three arms of the UPR. Overall, eighteen molecules, including the known ER stressors Tg and Tm, were identified to induce genes regulated by all three arms of the UPR, indicating that these molecules induce ER stress and/or global UPR activation (*Figure 2C*). In contrast, 79 compounds exhibited preferential activation of genes regulated in the ATF6 transcriptional program, relative to IRE1/XBP1s and PERK.

We prioritized 8 compounds for further characterization (*Table 1* — highlighted in *Figure 2C*). At this prioritization stage, we primarily focused on the MGE data and less emphasis was placed on the ATF6 or XBP1s luciferase reporter activation, as the MGE results provided richer information. The prioritized compounds were selected based on their ability to preferentially activate the ATF6 transcriptional program with varying magnitude according to the MGE analysis (*Figure 2C*). Furthermore, we focused on compounds containing a *2-amino-p-cresol* moiety (cluster F, *Figure 1—figure supplement 1B–C*), as these compounds were enriched within our top 79 preferential activators of the ATF6 transcriptional program (*Figure 2C*), but we ensured that other representatives of diverse structural classes were included. Only catechols (cluster A; *Figure 1—figure supplement 1B–C*) were mostly excluded, as these are known to be associated with adverse pharmacology (*Baell and Holloway, 2010*).

The toxicity of these 8 prioritized molecules was assessed in HEK293T-Rex cells. The compounds demonstrated maximum toxicity of <10–43% that of doxorubicin, with all of them exhibiting a cytotoxic concentration 50% ($CC_{50}$) of >17 μM (the highest tested concentration) (*Supplementary file 2*). These results suggest that our three-tiered screening strategy (*Figure 1A*) allowed identification of generally non-toxic small molecule ER proteostasis regulators that preferentially activate the ATF6 transcriptional program.

## mRNA-Seq and quantitative proteomics define the small molecule-mediated ER PN reprogramming

Comprehensive transcriptome analysis in HEK293T-Rex cells was employed to further validate the capacity of three prioritized small molecule ER proteostasis regulators — **132**, **147**, and **263** — to preferentially activate the ATF6 transcriptional program. These molecules were selected because they were structurally diverse and exhibited variable levels of ATF6 activation (measured by reporter assays and MGE) (*Table 1*, *Figure 2C*). As a global UPR activation control, mRNA-seq analysis was also performed on HEK293T-Rex cells treated with Tg.

This transcriptome analysis reveals that **132**, **147**, and **263** preferentially induce UPR-regulated genes involved in ER PN maintenance such as *BiP, GRP94* and *SEL1L* (*Figure 3—figure supplement 1A–C*, red). Compound **132** generated the most robust induction of these genes, relative to **147** and **263**, consistent with the levels of ERSE-FLuc activation afforded by these molecules (*Table 1*). Importantly, these small molecules induced three genes that were not upregulated by Tg-dependent global UPR activation — *HSPA1A, HSPA1B*, and *HMOX1* (*Figure 3—figure supplement 1D–F*). *HSPA1A* and *HSPA1B* are transcriptional targets of the heat-shock response (HSR)-associated transcription factor HSF1 (*Ryno et al., 2014*) and *HMOX1* is a transcriptional target of the oxidative stress response (*Yang et al., 2015*). Comparing the transcriptional profile previously observed for HSF1 activation (*Ryno et al., 2014*) to that observed for **132**, **147**, and **263** showed no significant correlation, indicating that these molecules do not induce the HSR (*Figure 3—figure supplement 1G–J*). Similarly, we did not observe increased expression of oxidative stress markers, indicating that these molecules do not globally activate the oxidative stress response (*Figure 3—figure supplement 1K*). These results confirm that our molecules selectively induce UPR-regulated ER proteostasis genes independent of other stress-responsive signaling pathways.

The ability of **132**, **147**, and **263** to preferentially activate transcriptional programs induced downstream of the ATF6, IRE1/XBP1s, and PERK arms of the UPR was next assessed from the mRNA-seq data. For this analysis, we defined genesets of >15 UPR target genes activated downstream of each of these three UPR signaling pathways. Previously published transcriptional profiles of chemical genetic activation of XBP1s, ATF6 or PERK defined these genesets (*Lu et al., 2004*; *Shoulders et al., 2013*). The induction of the selected genes by **132**, **147**, and **263** was then

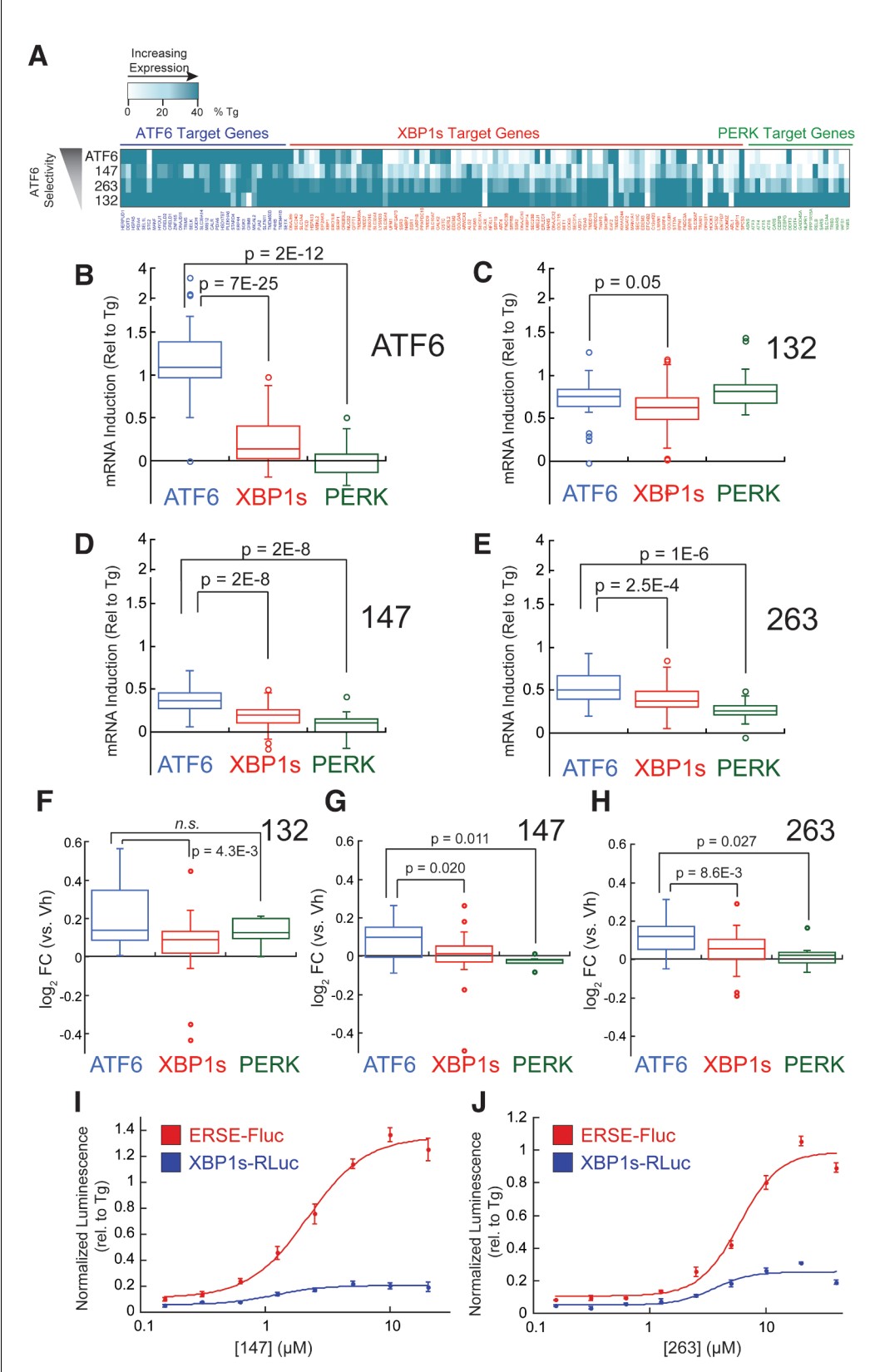

**Figure 3.** Small molecule ER proteostasis regulators preferentially activate the ATF6 transcriptional program. (**A**) Heat map showing the upregulation of ATF6 (blue), XBP1s (red) or PERK (green) target genes in the mRNA-Seq analysis of HEK293T-Rex cells treated with Tg (1 μM), **132** (10 μM), **147** (10 μM)
*Figure 3 continued on next page*

*Figure 3 continued*

or **263** (10 μM). Genes used in this analysis were selected from published reports indicating their selectivity for ATF6, XBP1s or PERK (*Lu et al., 2004*; *Shoulders et al., 2013*). The induction of each gene was normalized to the respective induction observed with the global UPR activator Tg and reported as % Tg induction. The expression of these target genes in the mRNA-Seq analysis of TMP-dependent DHFR-ATF6 activation in HEK293^DAX cells is also shown. The data used to prepare this heat map is included in *Figure 3—source data 1*. (B–E) Box plots showing the relative activation of ATF6, XBP1s, and PERK genesets in HEK293^DAX cells following TMP-dependent DHFR-ATF6 activation (B), or HEK293T-Rex cells treated with **132** (C), **147** (D) or **263** (E) from the data shown in *Figure 3A*. Differences in activation of the ATF6 geneset relative to the XBP1s or PERK genesets were confirmed by one-way ANOVA and the *p*-values of unpaired t-tests are shown. (F–H) Box plots comparing the average fold change at the protein levels of ATF6, XBP1s and PERK target sets in HEK293T-Rex cells treated with 10 μM **132** (F), **147** (G), or **263** (H) for 16 hr as observed in the proteomics analysis. The protein sets used here are the same as the genesets for the analysis of the RNA-Seq data (A–E). Differences in activation of the ATF6 target protein set relative to the XBP1s or PERK target protein sets were confirmed by one-way ANOVA and the p-values of unpaired t-tests are shown. The data used to prepare these graphs can be found in *Figure 3—source data 1*. (I–J) Plot showing the relative activation of the ERSE-FLuc (blue) and XBP1-RLuc (red) reporters in HEK293T-Rex cells treated with the indicated concentration of **147** (I) or **263** (J) for 15 hr. Error bars represent standard error for n = 3 replicates.

The following source data and figure supplements are available for figure 3:

**Source data 1.** Excel spreadsheet containing the RNA-seq and proteomic geneset data used to prepare *Figure 3A–H*.

**Figure supplement 1.** mRNA-Seq analysis shows that ER proteostasis regulators do not globally activate the Heat Shock Response (HSR) or oxidative stress response pathways.

**Figure supplement 1—source data 1.** Excel spreadsheet describing the RNA-seq data used to prepare *Figure 3—figure supplement 1A–K*.

**Figure supplement 2.** ER proteostasis regulators induce ATF6 targets at the protein level and time-dependence of ATF6 and XBP1s target activation.

**Figure supplement 2—source data 2.** Excel spreadsheet describing the whole cell proteomic data used to prepare *Figure 3—figure supplement 2A–C*.

normalized to that observed with Tg, allowing direct comparisons across the three genesets (*Figure 3—source data 1*). The ability of this approach to define preferential ER proteostasis remodeling was confirmed by demonstrating that chemical genetic ATF6 activation in HEK293^DAX cells selectively induces the ATF6 transcriptional program (*Figure 3A,B*).

Compound **132** significantly activates the ATF6 transcriptional program (75% relative to Tg), but also robustly induces expression of IRE1/XBP1s and PERK genesets (*Figure 3A,C*). While modest preferential activation of the ATF6 transcriptional program is observed, this indicates that **132** is a global UPR activator. In contrast, compound **147** and to a lesser extent **263** show preferential activation of the ATF6 transcriptional program relative to IRE1/XBP1s and PERK albeit to lower levels, showing 45% and 50% activation relative to Tg, respectively (*Figure 3A,D,E*).

Reprogramming of the ER PN was next monitored using quantitative proteomics in HEK293T-Rex cells treated with **132**, **147** or **263**. Protein level changes induced by the addition of these small molecules correlated with transcript changes observed by mRNA-seq (*Figure 3—figure supplement 2A–C*). Furthermore, the extent of protein expression changes followed the same trend as the transcript changes: induction was highest with compound **132**, relative to **263** and **147**. We further confirmed this trend by quantitative immunoblotting monitoring the expression of several proteins regulated by the ATF6 transcriptional program (*Figure 3—figure supplement 2D*). Applying an analogous geneset analysis to that described above showed that **132** increased levels of proteins regulated downstream of ATF6, IRE1/XBP1s and PERK to similar extents, further confirming that **132** is a global UPR activator (*Figure 3F*). However, **147** and **263** preferentially increased levels of proteins regulated by the ATF6 transcriptional program, further confirming that these molecules reprogram the ER PN through a mechanism involving preferential activation of this transcriptional program (*Figure 3G,H*).

Preferential activation of the ATF6 transcriptional program by **147** and **263** could result from a mild level of ER stress that should increase upon addition of higher concentrations of these molecules or longer duration of treatment. When monitoring the activation of the ATF6-selective ERSE-FLuc reporter in HEK293 cells treated with increasing concentrations of **147** and **263** (*Figure 3I,J*),

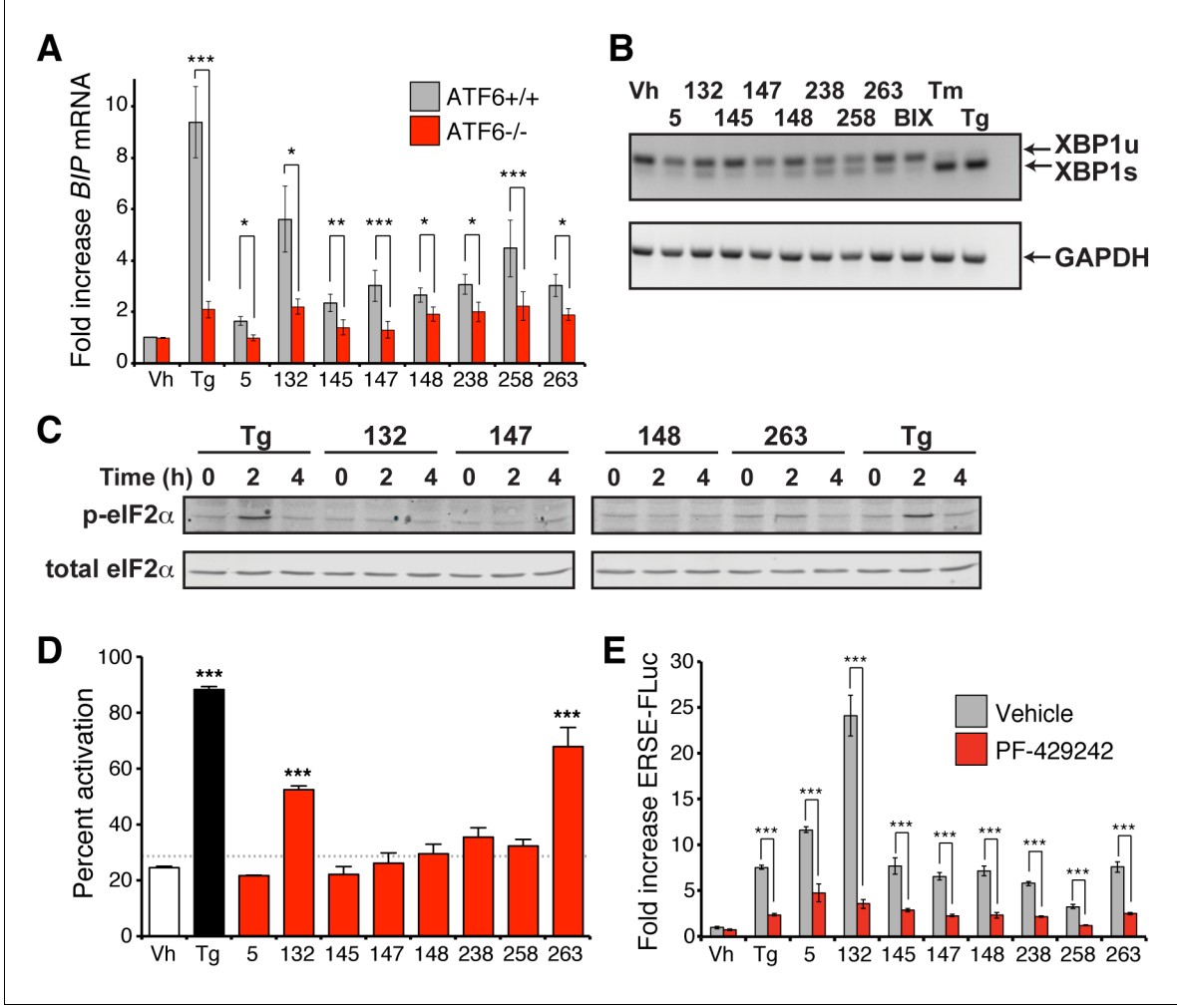

**Figure 4.** ER proteostasis regulators depend on endogenous ATF6 activation. (**A**) *BIP* mRNA measured by qPCR in ATF6^{+/+} and ATF6^{-/-} MEFs treated with the indicated small molecule ER proteostasis regulators (10 μM; 6 hr). Error bars show standard error for n > 3. *p<0.05; **p<0.01; ***p<0.001. (**B**) Gel showing XBP1 splicing in HEK293T cells measured by RT-PCR. RNA was extracted from cells treated with global ER stressors Tg (500 nM) or tunicamycin (Tm; 1 μg/mL) or 10 μM of the indicated small molecule ER proteostasis regulator for 3 hr. The small molecule BIX is included as a control. After generation of cDNA, primers flanking the XBP1 splicing site were used to amplify the longer unspliced (XBP1u) or 26-nt shorter spliced (XBP1s) segment. (**C**) Immunoblot showing eIF2α phosphorylation in HEK293T cells treated with Tg (500 nM) or 10 μM of the indicated small molecule ER proteostasis regulator for the indicated time. (**D**) Activation of ATF6 as measured by nuclear localization of GFP-ATF6. U2OS-GFP-ATF6 cells were treated with the top 8 small molecule ER proteostasis regulators (10 μM; 5 hr) or Tg (100 nM; 5 hr) and subcellular localization of GFP was assessed by confocal microscopy (representative images are shown in *Figure 4—figure supplement 1A*). The nuclear:ER ratio of GFP signal corresponding to activation of GFP-ATF6 was calculated by comparing vehicle to Tg treatment. Error bars show standard error for n = 3 replicates. Dotted line shows mean plus three standard deviations from vehicle treated controls (28.65%). ***p<0.001 using one way ANOVA followed by Tukey's Multiple comparison test (compared to vehicle). (**E**) Plot showing ERSE-FLuc activation in HEK293T-Rex cells treated with the top 8 small molecule ER proteostasis regulators (10 μM; 18 hr) in the presence (red bars) or absence (grey bars) of the S1P inhibitor PF-429242 (10 μM; 18 hr). Cells treated with Tg (500 nM; 18 hr) are shown as a control. Error bars show standard error for n = 3. ***p<0.001.

The following figure supplement is available for figure 4:

**Figure supplement 1.** Select ER proteostasis regulators show proteolytic processing and nuclear translocation of ATF6.

neither compound showed significant increases in ERSE-FLuc activation at concentrations >10 μM – the concentration used for mRNA-seq. Similarly, increasing the concentration of these molecules did not significantly increase activation of the IRE1/XBP1s-selective XBP1-RLuc reporter. Furthermore, we observe a maximal *BiP* induction at 6 hr in cells treated with **147** or **263** over a 24 hr time course and no time-dependent increases in expression of the XBP1s target *Sec24D* (*Figure 3—figure*

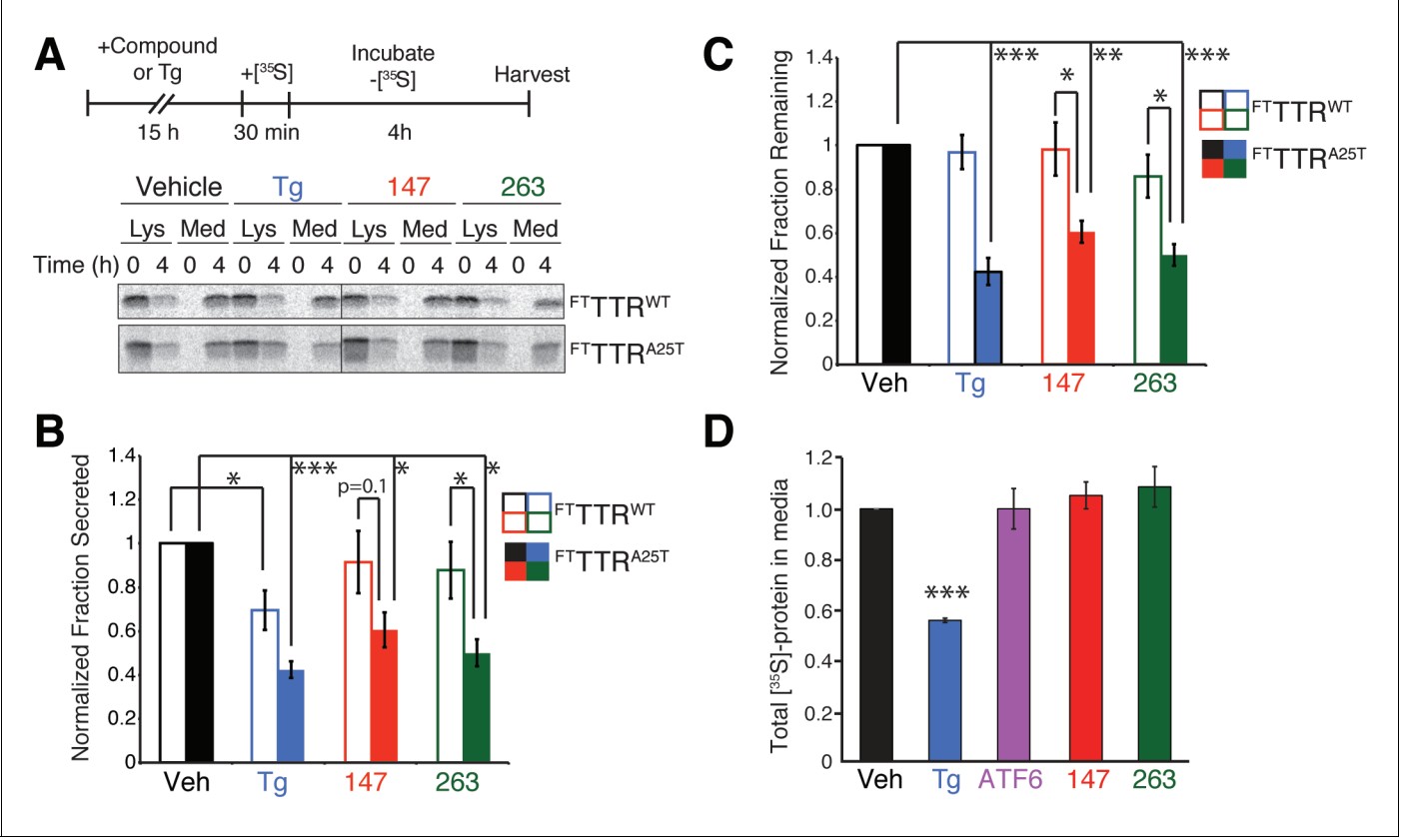

**Figure 5.** Small molecule ER proteostasis regulators reduce secretion of destabilized amyloidogenic $^{FT}$TTR$^{A25T}$ from HepG2 cells. (**A**) Representative autoradiogram of [$^{35}$S]-labeled $^{FT}$TTR$^{WT}$ or $^{FT}$TTR$^{A25T}$ in the lysate (Lys) and media (Med) of HepG2$^{ATF6}$ cells pretreated for 15 hr with 10 μM small molecule **147** or **263**, or 1 μM Tg, incubated with [$^{35}$S] for 30 min, and then incubated in fresh non-radioactive media for 0 or 4 hr. The experimental paradigm is shown above. (**B**) Quantification of the normalized fraction of $^{FT}$TTR$^{WT}$ (open bars) or $^{FT}$TTR$^{A25T}$ (colored bars) secreted following a 4 hr chase in HepG2$^{ATF6}$ cells pretreated for 15 hr with Tg (1 μM; blue) or 10 μM small molecule **147** (red) or **263** (green). Fraction secreted was calculated using the equation: fraction secreted = [extracellular [$^{35}$S]-TTR signal at t=4h / (extracellular [$^{35}$S]-TTR signal at t=0h + intracellular [$^{35}$S]-TTR signal at t=0h)]. Normalized fraction secreted was calculated relative to vehicle-treated controls using the equation: normalized fraction secreted = fraction secreted treatment / fraction secreted vehicle. Representative autoradiograms are shown in *Figure 5A*. Raw fraction secreted is shown in *Figure 5— figure supplement 1C*. Error bars show standard error for n = 4 replicates; *p<0.05, ***p<0.001. (**C**) Quantification of the normalized fraction of $^{FT}$TTR$^{WT}$ (open bars) or $^{FT}$TTR$^{A25T}$ (colored bars) remaining (lysate + media) following a 4 hr chase in HepG2$^{ATF6}$ cells pretreated for 15 hr with Tg (1 μM; blue) or 10 μM small molecule **147** (red) or **263** (green). Fraction remaining was calculated using the equation: [(extracellular [$^{35}$S]-TTR signal at t=4h + intracellular [$^{35}$S]-TTR signal at t=4h) / (extracellular [$^{35}$S]-TTR signal at t=0h + intracellular [$^{35}$S]-TTR signal at t=0h)]. Normalized fraction remaining was calculated relative to vehicle-treated controls using the equation: normalized fraction remaining = fraction remaining treatment / fraction remaining vehicle. Representative autoradiograms are shown in *Figure 5A*. Error bars show standard error for n = 4 replicates; *p<0.05, **p<0.01, ***p<0.001. (**D**) Quantification for normalized total [$^{35}$S]-labeled proteins secreted from HepG2$^{ATF6}$ cells following 15 hr pretreatment with 1 μM Tg (blue), TMP (10 μM)-dependent DHFR-ATF6 activation (purple), or 10 μM small molecule ER proteostasis regulators **147** (red) or **263** (green). The experimental protocol and a representative autoradiogram for this experiment is shown in *Figure 5—figure supplement 1F*. The recovery of [$^{35}$S] labeled proteins is normalized to the vehicle-treated control. Error bars show standard error for n = 4 replicates. ***indicates p<0.001.

The following figure supplement is available for figure 5:

**Figure supplement 1.** ER proteostasis regulators reduce secretion of destabilized amyloidogenic TTR$^{A25T}$ from HepG2 cells.

---

*supplement 2E–F*). These results indicate that increasing the concentration or duration of treatment with small molecule ER proteostasis regulators does not alter their preferential activation of the ATF6 transcriptional program, suggesting that the compounds do not function by inducing a mild level of ER stress.

## Small molecule activation of the ATF6 transcriptional program involves processing of endogenous ATF6

We next evaluated whether the preferential activation of the ATF6 transcriptional program afforded by our small molecules requires endogenous ATF6. Expression of the ATF6 target gene *BiP* was measured in ATF6$^{+/+}$ mouse embryonic fibroblast (MEF) cells treated with the top 8 prioritized small molecule ER proteostasis regulators shown in *Table 1*. All of these molecules induced *BiP* expression in these cells, albeit to varying extents (*Figure 4A*). In contrast, *BiP* induction was significantly inhibited in the ATF6$^{-/-}$ MEFs, showing that small molecule-dependent upregulation of a gene included in the ATF6 transcriptional program requires endogenous ATF6. Importantly, we did not observe significant levels of *XBP1* splicing or eIF2α phosphorylation in cells treated with the ER proteostasis regulators, indicating that these molecules do not significantly activate the IRE1/XBP1s or PERK arms of the UPR (*Figure 4B,C*).

ATF6 activation involves trafficking of the full-length ATF6 protein to the Golgi. ATF6 is then processed by site-1 and site-2 proteases (S1P/S2P) to release the active cytosolic bZIP transcription factor, facilitating nuclear localization (*Ye et al., 2000*). To evaluate the influence of the 8 ER proteostasis regulators on ATF6 processing, we measured ATF6 nuclear translocation using a fluorescence-based high-content imaging assay that quantifies nuclear-localized ATF6-GFP relative to ER-localized ATF6-GFP. Treatment with small molecules **132** and **263** increased the nuclear fraction of ATF6-GFP, reflecting increased ATF6 processing (*Figure 4D—figure supplement 1A*). Compound **132** and **263** treatment also led to increased accumulation of the 50 kDa ATF6 fragment, resulting from S1P/S2P endoproteolysis in the Golgi, as shown by immunoblotting (*Figure 4—figure supplement 1B*). Other molecules, including **147**, did not significantly increase ATF6-GFP nuclear localization or accumulation of the 50 kDa ATF6 fragment, although it cannot be excluded that they also induce ATF6 processing, albeit to a lower extent that falls outside the dynamic range or sensitivity of these assays.

To more sensitively evaluate the involvement of ATF6 endoproteolytic processing in small molecule-dependent activation of the ATF6 transcriptional program, we utilized the S1P inhibitor PF-429242, which prevents Tg-mediated ERSE-FLuc activation (*Figure 4E*) (*Hay et al., 2007*). PF-429242 also inhibited the activation of ERSE-FLuc in cells treated with all 8 ER proteostasis regulators (*Figure 4E*), reflecting the dependence of our small molecules on S1P proteolysis for ATF6 activation. Collectively, the ATF6 knockout, nuclear localization and S1P inhibition results indicate that the small molecule ER proteostasis regulators activate the ATF6 transcriptional program through a mechanism involving S1P-dependent processing of endogenous ATF6.

## ER proteostasis regulators selectively reduce secretion of amyloidogenic TTR from liver-derived HepG2 cells

ER reprogramming afforded by stress-independent chemical genetic ATF6 activation selectively reduces secretion of destabilized, amyloidogenic TTR variants from liver-derived HepG2 cells (*Chen et al., 2014*; *Shoulders et al., 2013*). Thus, small molecule ER proteostasis regulators that preferentially activate the ATF6 transcriptional program, such as **147** and **263**, should similarly reduce secretion of destabilized, amyloidogenic TTR from HepG2 cells. We confirmed that **147** and **263** induced expression of the ATF6-target gene *BiP* in HepG2 cells by qPCR (*Figure 5—figure supplement 1A*). We also showed that the treatment dose of these small molecules (10 µM) does not influence HepG2 viability, although higher concentrations (100 µM) do reduce viability (*Figure 5—figure supplement 1B*).

We used [$^{35}$S] metabolic labeling to quantify secretion of FLAG-tagged destabilized A25T TTR ($^{FT}$TTR$^{A25T}$) or FLAG-tagged wild-type TTR ($^{FT}$TTR$^{WT}$) from HepG2 cells stably expressing DHFR-ATF6 (HepG2$^{ATF6}$). Cells were pretreated with the global ER stressor Tg or small molecule ER proteostasis regulators **147** or **263** (*Figure 5A*). Pretreatment with Tg reduced secretion of both $^{FT}$TTR$^{A25T}$ and $^{FT}$TTR$^{WT}$ (*Figure 5B—figure supplement 1C*). However, pretreatment with **147** and **263** reduced $^{FT}$TTR$^{A25T}$ secretion by 45%, but did not significantly influence secretion of $^{FT}$TTR$^{WT}$ (*Figure 5B—figure supplement 1C*). The reduction in $^{FT}$TTR$^{A25T}$ secretion induced by these molecules correlates with increased degradation of $^{FT}$TTR$^{A25T}$ (*Figure 5C*). Notably, the selective reduction in secretion and increase in degradation for $^{FT}$TTR$^{A25T}$ induced by **147** and **263** is strictly analogous to that observed after chemical genetic ATF6 activation in these cells (*Figure 5—figure*

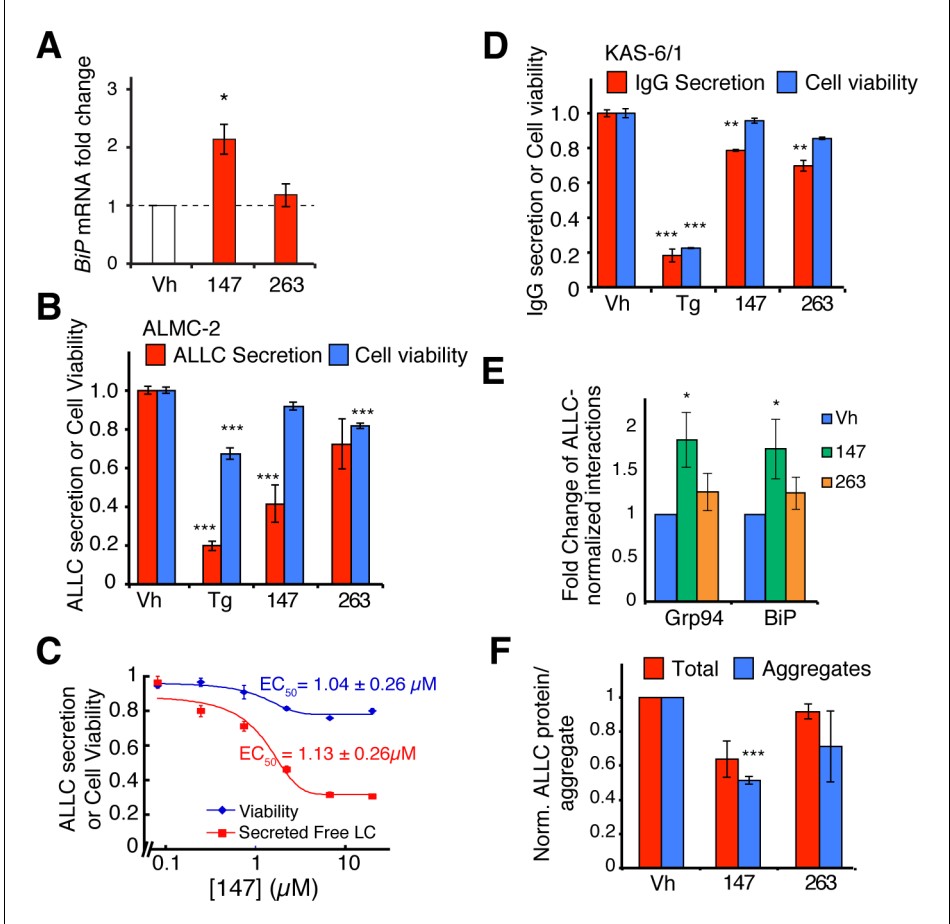

**Figure 6.** Small molecule ER proteostasis regulators reduce secretion of destabilized, amyloidogenic immunoglobulin light chains from AL patient-derived plasma cells. (A) qPCR analysis of *BiP* expression in ALMC-2 cells treated for 6 hr with 10 µM of the indicated small molecule ER proteostasis regulators. Error bars show standard error for n = 3 replicates. *p< 0.05. (B) Graph showing relative media levels of ALLC measured by ELISA (red) and cellular viability (blue) of ALMC-2 cells pretreated with 10 µM of the indicated small molecule ER proteostasis regulators. Cells treated with 500 nM Tg are shown as a control. Error bars show standard error for n > 6 replicates. ***p<0.001. (C) Dose response curves of reduction in ALLC secretion from ALMC-2 cells after **147** treatment. ALMC-2 cells were pretreated for 15 hr, and subsequently the media was exchanged and conditioned on the cells for 2 hr. Secreted ALLC concentration was measured by ELISA using an antibody directed against free λ LC (red). Cell viability was measured by CellTiter-Glo (blue). Error bars represent standard error for n = 3 replicates. Data was fit to four parameter logistic functions to calculate $EC_{50}$ values. (D) Graph showing relative media levels of IgG measured by ELISA (red) and cellular viability (blue) of KAS-6/1 cells treated with 10 µM of the indicated small molecule ER proteostasis regulators. Cells treated with 500 nM Tg are shown as a control. Error bars show standard error for n = 3 replicates. **p<0.01, ***p<0.001. (E) Quantification of ALLC immunopurifications from ALMC-2 cells after treatment with the indicated ER proteostasis regulator (10 µM; 16 h) and in situ cross-linking. The relative recovery of GRP94 and BiP in each sample normalized to the recovered ALLC is shown. A representative immunoblot is shown in *Figure 6—figure supplement 1D*. Error bars show standard error for n = 4 replicates. *p<0.05. (F) Quantification of ALLC aggregates in conditioned media prepared on ALMC-2 cells pretreated with the indicated ER proteostasis regulator (10 µM; 16 hr). After pretreatment, media were exchanged and the volume adjusted to account for decreases in cell viability with several compounds. After conditioning for 8 hr, the media were collected and then heated to 55°C for 8 hr to induce ALLC aggregation. Quantification of the total (red) and aggregate (blue) ALLC amounts are shown. Error bars show standard error for n > 3 replicates. *p<0.05, **p<0.01, ***p<0.001. Blue Native (BN) PAGE/Immunoblots of the conditioned media are shown in *Figure 6—figure supplement 1H*. Quantification of ALLC aggregates in conditioned media pretreated with additional ER proteostasis regulator compounds is shown in *Figure 6—figure supplement 1I*.

The following figure supplement is available for figure 6:

**Figure supplement 1.** ER proteostasis regulators reduce secretion and extracellular aggregation of a destabilized, amyloidogenic light chain from AL patient-derived plasma cells.

*supplement 1D,E*) (*Chen et al., 2014*). This indicates that these small molecule ER proteostasis regulators phenocopy the capacity of chemical genetic ATF6 activation to selectively reduce secretion of destabilized, amyloidogenic TTR variants relative to the wild-type protein.

We next evaluated whether pretreatment by **147** and **263** influences secretion of the endogenous secretory proteome from HepG2^ATF6 cells using a [$^{35}$S] metabolic labeling approach (*Figure 5—figure supplement 1F*) (*Chen et al., 2014*; *Shoulders et al., 2013*). Treatment with the ER stressor Tg significantly reduced the recovery of [$^{35}$S] labeled proteins in conditioned media, consistent with the ability of ER stress to induce translational attenuation and globally disrupt protein secretion (*Figure 5D—figure supplement 1F*). However, chemical genetic ATF6 activation, or pharmacologic activation of the ATF6 transcriptional program through addition of compound **147** or **263**, did not influence the levels of [$^{35}$S] labeled proteins in media, demonstrating that preferential ATF6 activation does not globally disrupt secretion of the endogenous proteome.

## ER proteostasis Rregulators Ddecrease Ssecretion and extracellular aggregation of an Aamyloidogenic Llight chain

We next tested whether the ER proteostasis regulators reduce the secretion of a destabilized, amyloidogenic immunoglobulin LC (ALLC) that undergoes proteotoxic extracellular aggregation in association with AL (*Arendt et al., 2008*). Chemical genetic ATF6 activation reduces secretion and extracellular aggregation of ALLC without notably affecting secretion of a stable, non-amyloidogenic LC (*Cooley et al., 2014*). We evaluated whether our prioritized ER proteostasis regulators similarly reduce secretion of the same ALLC from an AL patient-derived plasma cell line (ALMC-2) (*Arendt et al., 2008*). Compound **147** induced expression of the ATF6 target gene *BiP* in ALMC-2 cells but unexpectedly, **263** did not (*Figure 6A*). Treatment with compound **147** reduced ALLC secretion >50% from ALMC-2 cells, as measured by an ELISA while **263** did not significantly influence ALLC secretion (*Figure 6B*). Dose-dependent treatment with **147** measuring ALLC secretion afforded an EC$_{50}$ value of 1.1 µM (*Figure 6C*). The attenuation in ALLC secretion saturated above 10 µM, producing a similar dose-response curve to **147**-dependent ATF6 reporter activation (*Figure 3I*). Conversely, compared to the reduction in ALLC secretion, these molecules only minimally affected the secretion of an energetically normal, fully assembled IgG from a control KAS-6/1 plasma cell line isolated from a multiple myeloma patient that did not present with AL (*Figure 6D*) (*Westendorf et al., 1996*). These data indicate that small molecule ER proteostasis regulators that preferentially activate the ATF6 transcriptional program, e.g., **147**, selectively reduce secretion of destabilized, amyloidogenic LCs, but not IgGs. Additional ER proteostasis regulator compounds tested also reduced secretion of amyloidogenic ALLC from ALMC-2 cells, while only minimally affecting IgG secretion from KAS-6/1 cells (*Figure 6—figure supplement 1A,B*).

Tg treatment reduced viability of ALMC-2 and KAS-6/1 cells (*Figure 6B,D*). Thus, reductions of ALLC secretion afforded by these treatments in part reflect cell death. However, the preferential activator of the ATF6 transcriptional program, **147**, only minimally influenced the viability of ALMC-2 or KAS-6/1 cells (*Figure 6B–D*), suggesting that the reductions in ALLC secretion afforded by this molecule involve reprogramming of the ER PN. Consistent with this prediction, small molecule-dependent reductions in ALLC secretion from ALMC-2 cells strongly correlate with *BiP* induction in these cells (*Figure 6—figure supplement 1C*). Furthermore, immunopurification of ALLC from ALMC-2 cells pretreated with **147** and subjected to in situ crosslinking showed increased interactions between ALLC and the ATF6-regulated chaperones BiP and GRP94 (*Figure 6E*, *Figure 6—figure supplement 1D*). However, pretreatment with **263**, which did not induce *BiP* in ALMC-2 cells, did not increase the co-isolation of these chaperones. These results suggest that pharmacologic reductions in ALLC secretion are linked to reprogramming of the ER proteostasis environment.

We previously showed that chemical genetic ATF6 activation reduces ALLC secretion through a mechanism involving prolonged ER retention (*Cooley et al., 2014*). In contrast, genetic XBP1s activation reduces ALLC secretion by increasing its degradation. Therefore, evaluating the mechanism by which the small molecules reduce ALLC secretion provides insight into the specific type of ER reprogramming involved in this process. A cycloheximide chase assay demonstrated that pretreatment with **147** reduced ALLC secretion by 30% through a mechanism involving increased ER retention (*Figure 6—figure supplement 1E–G*). This indicates that **147** reduces secretion of ALLC through a mechanism similar or identical to that previously observed with chemical genetic ATF6 activation.

Finally, we evaluated whether pharmacologic ATF6-mediated reductions in ALLC secretion attenuated extracellular ALLC aggregation. Soluble ALLC aggregates were quantified in conditioned media from ALMC-2 cells after heating to 55°C for 8 hr using Blue-Native PAGE and immunoblotting (*Cooley et al., 2014*). Total ALLC protein levels were also determined by SDS-PAGE and immunoblotting. Treatment with **147** decreased extracellular ALLC aggregates by >50% (*Figure 6F—figure supplement 1H*). Similar reduction in aggregation was observed for additional ER proteostasis regulators tested that lowered ALLC secretion (*Figure 6F—figure supplement 1H,I*). This indicates that the reduction in ALLC secretion afforded by our ER proteostasis regulators attenuates extracellular LC aggregation.

## Discussion

Reprogramming of subcellular proteostasis environments through activation of stress-responsive signaling pathways is a promising strategy to restore mutant protein folding or enhance mutant protein degradation (*Calamini and Morimoto, 2012*; *Chen et al., 2015*; *Das et al., 2015*). Here, we employed high-throughput screening in combination with medium-throughput transcriptional profiling to identify small molecule ER proteostasis regulators that preferentially activate the ATF6 transcriptional program of the UPR. We show that pharmacologic activation of the ATF6 transcriptional program requires S1P-dependent processing of endogenous ATF6. Furthermore, we show that our small molecule ER proteostasis regulators phenocopy the capacity of chemical genetic ATF6 activation to selectively reduce secretion and extracellular aggregation of destabilized amyloidogenic variants of proteins such as TTR and LC. These results reveal a new strategy to identify small molecules that reprogram the ER PN and correct imbalances in ER function through preferential activation of a specific sub-UPR transcriptional program.

Our phenotypic screening approach sought to identify small molecule ER proteostasis regulators that activated the ATF6 transcriptional program, without placing a constraint for the compounds to act directly on ATF6. Nonetheless, prioritized compounds such as **147** and **263** function through a mechanism that requires endogenous ATF6. ATF6 is activated through a mechanism distinct to that utilized by the two other arms of the UPR. In response to ER stress, ATF6 traffics to the Golgi where it is endoproteolytically processed by S1P and S2P (*Ye et al., 2000*). This releases the ATF6 bZIP transcription factor into the cytosol, facilitating its translocation to the nucleus where it promotes induction of the ATF6 transcriptional program. ATF6 is not known to contain any allosteric small molecule binding sites. Thus, it is likely that our molecules promote ATF6 activation through interactions with proteins that regulate its activity such as BiP and the PDIs. Both of these are predicted to be involved in initiating ATF6 trafficking to the Golgi, although their role in this process remains poorly defined (*Nadanaka et al., 2007*; *Shen et al., 2002*). Our small molecules could also preferentially induce ATF6 activity by targeting or mimicking metabolites required for activation of this pathway. Previous work shows that PERK and IRE1 can be activated independent of ATF6 in response to alterations in cellular lipid composition (*Volmer et al., 2013*) but no such mechanism has been reported for ATF6 to date.

Interestingly, ER proteostasis regulators identified in our screen appear to preferentially induce ATF6 activation through distinct mechanisms. This is evident for molecules such as **263**, which demonstrates varying levels of *BiP* induction in distinct cell types (e.g., **263** induces *BiP* in HEK293T and HepG2 cells, but not in ALMC-2 cells). This is in contrast to compound **147** that activates the ATF6 pathway in all cell types tested. This capacity to activate ATF6 in a cell type specific fashion could be harnessed to define the underlying molecular mechanisms involved in ATF6 activation and develop compounds that induce tissue-selective ER PN remodeling. As we continue to optimize the potency and ATF6 selectivity of these small molecule ER proteostasis regulators, new insights into the molecular pathways that regulate ATF6 activity will likely emerge.

One mechanism by which the ATF6 transcriptional program influences ER function is by increasing the stringency of ER protein quality control (*Chen et al., 2015*). Stress-independent ATF6 activation selectively attenuates the secretion and subsequent proteotoxic aggregation of destabilized, disease-associated proteins without affecting the secretion of the endogenous proteome (*Chen et al., 2014*; *Chiang et al., 2012*; *Cooley et al., 2014*; *Shoulders et al., 2013*; *Smith et al., 2011*). Our top small molecule ER proteostasis regulators (e.g., **147**) phenocopy the ability of chemical genetic ATF6 activation to selectively reduce secretion of destabilized, disease-associated variants of

amyloidogenic proteins such as TTR and immunoglobulin LC. This suggests that our small molecules could similarly phenocopy the ATF6-dependent reductions in the trafficking and intracellular aggregation of destabilized, aggregation-prone proteins associated with other gain-of-toxicity protein misfolding diseases (e.g., A1AT-Z aggregation in liver disease and rhodopsin aggregation in retinal degeneration (*Chiang et al., 2012*; *Smith et al., 2011*)). Thus, small molecule ER proteostasis regulators identified herein have the potential to broadly ameliorate degenerative diseases associated with aggregation of secretory proteins.

The long-term consequences of periodic pharmacologic activation of the ATF6 arm of the UPR are unclear. Mechanism-based toxicity associated with adapting ER proteostasis to ameliorate secretory protein misfolding and/or aggregation is possible. While some small molecule ER proteostasis regulators identified show mild toxicity, a subset of compounds, especially compounds that preferentially activate the ATF6 transcriptional program (e.g., **147**), show no significant toxicity across multiple cell models. This likely reflects the limited activation of the IRE1/XBP1s and PERK arms of the UPR afforded by these molecules, minimizing apoptosis signaling induced downstream of these two UPR signaling pathways (*Hetz, 2012*; *Tabas and Ron, 2011*). The moderate levels of ATF6 activation achieved using our small molecule ER proteostasis regulators likely also minimize pathologic imbalances in ER function. It is known that high levels of ATF6 activation induce hepatic steatosis in zebrafish, whereas lower levels do not (*Howarth et al., 2014*). Finally, the low molecular weight and structural simplicity of the small molecules identified by our multi-tiered screening strategy (e.g., **147**) provides a significant amount of chemical space that can be explored through structure-activity relationships to optimize potency, selectivity and dosing regimens, while minimizing toxicity.

Several of our prioritized molecules include functional groups found in pan-assay interference compounds (PAINS) (*Baell and Walters, 2014*), but given that these ER proteostasis regulators showed activity in multiple complementary screening assays, they are unlikely to act through off-target pathways. Consistent with this, the molecules depend on ATF6 and its canonical activation pathway for the induction of ATF6 target genes. Furthermore, the RNA-Seq and proteomics profiling analysis for **147** and **263** confirmed that these molecules exert very little off-target transcriptome changes and proteome remodeling apart from preferential ER PN activation. Both **147** and **263** contain functional groups that after metabolic oxidation to a quinone methide could modify proteins covalently. This suggests that the covalent modification of ATF6 or a protein involved in the regulation of ATF6 activity (e.g., BiP or PDIs) (*Nadanaka et al., 2007*; *Shen et al., 2002*) by our small molecules could be involved in the activation mechanism. Future medicinal chemistry efforts will focus on exploring the mechanism of action of these compounds and the potential involvement of covalent protein modification for their activity.

The establishment of ER proteostasis regulators, such as **147**, provides a unique opportunity to define the underlying molecular mechanism(s) of ATF6 activation and to probe the involvement of ATF6 in regulating ER PN function. These molecules also represent a valuable resource to define the therapeutic potential of ATF6 transcriptional program activation to correct pathologic imbalances in ER proteostasis in cellular and animal models of protein misfolding and aggregation diseases. As we continue to develop our top small molecule ER proteostasis regulators through medicinal chemistry, we will establish second generation molecules that optimize ER PN reprogramming to ameliorate pathologic imbalances in ER proteostasis associated with diverse protein aggregation diseases such as the systemic amyloidosis.

## Materials and methods

### Compounds

All compounds were obtained from the commercial vendors listed in *Supplementary file 3* and were used without further purification by dissolving in sterile dimethyl sulfoxide (DMSO). The identity of screening compounds was confirmed by LC-MS analysis and all compounds were shown to be >90% pure.

### Plasmids

ERSE-Firefly luciferase reporter was cloned into a vector suitable for mammalian cell selection by transferring ERSE-FLuc from ERSE-FLuc.pGL3 (*Yoshida et al., 1998*) into a promotorless pcDNA3.1

vector using Xba1 and Not1 restriction sites to create ERSE.FLuc.pcDNA3.1. XBP1s-Renilla luciferase was generated from a known XBP1s-GFP reporter (*Iwawaki et al., 2004*). GFP was exchanged for Renilla luciferase by the Polymerase Incomplete Primer Extension (PIPE) method (*Klock and Lesley, 2009*) using the following primers: vector, XBP1s-GFP.pCEFL, CTGAAGAACGAGCAGTAAGTGAG-CAAGGGCGAGGAG and CGTACACCTTGGAAGCAGATCTTGAATCTGAAGAGTCAATACC; gene, CMV-Renilla, CGGTATTGACTCTTCAGATTCAAGATCTGCTTCCAAGGTGTACG and CTCCTCGCCC TTGCTCACTTACTGCTCGTTCTTCAG to create XBP1s-RLuc pCEFL. TTR point mutations were incorporated into the FLAG$_2$-TTR pcDNA3.1 vector through site-directed mutagenesis (*Shoulders et al., 2013*).

## Cell culture and transfections

HEK293T-Rex (ATCC), HEK293T (ATCC), HEK293$^{DAX}$ (*Shoulders et al., 2013*), HepG2 (ATCC), HepG2$^{ATF6}$(*Chen et al., 2014*), and ATF6$^{+/+}$ MEFs and ATF6$^{-/-}$ MEFs (a kind gift from Randal Kaufman's lab) were cultured in High-Glucose Dulbecco's Modified Eagle's Medium (DMEM) supplemented with glutamine, penicillin/streptomycin and 10% fetal bovine serum. Cells were routinely tested for mycoplasma every 6 months. No further authentication of cell lines was performed by the authors. HEK293T-Rex cells containing the ERSE-FLuc or XBP1s-RLuc reporters were created by transfection with ERSE-FLuc pcDNA3.1 or XBP1s-RLuc pCEFL by calcium phosphate followed by culturing in geneticin sulfate (G-418, 500 µg/mL, ERSE) or puromycin (20 µg/mL, XPB1s). Creation and maintenance of HEK293$^{DAX}$ cells has been described previously (*Shoulders et al., 2013*). Creation and maintenance of HepG2$^{ATF6}$ has been described previously (*Chen et al., 2014*). U2OS cells stably expressing GFP-ATF6 were purchased from Thermo Fisher Scientific (Waltham, MA) (084_01) and cultured with 500 µg/mL G418. Transient transfections of $^{FT}$TTR variants into HepG2$^{ATF6}$ cells were performed with Lipofectamine 3000 (Life Technologies, Carlsbad, CA). ALMC-2 and KAS-6/1 plasma cells (a kind gift from Diane Jelinak's laboratory) were cultured in Iscove's Modified Dulbecco's Medium (IMDM) GlutaMAX (Life Technologies) supplemented with penicillin/streptomycin, 5% fetal bovine serum and 2 ng/mL interleukin-6 (IL-6), as previously described (*Arendt et al., 2008*). All cells were cultured under typical tissue culture conditions (37°C, 5% CO$_2$).

## Reporter assays

### 96-well, HEK293$^{DAX}$ experiments and S1P inhibition

HEK293T-Rex cells incorporating the ERSE-FLuc reporter and HEK293$^{DAX}$ cells were plated at approximately 20,000 cells/well in flat-bottomed, black 96-well assay plates (Costar, Corning, Inc., Corning, NY) overnight prior to compound administration. Cells were treated with compounds as described for 12–18 hr, then the plates were equilibrated to room temperature and 50 µL of SteadyLite (PerkinElmer, Waltham, MA) was added to each well. Luminescence activity was measured in a Safire II microplate reader with a 1000 ms integration time.

### 384-well, Tg and Tm dose response in reporter cells

HEK293T-Rex cells incorporating either the ERSE-FLuc or XBP1s-RLuc reporters were plated at 20 µL/well from 250,000 cells/mL in white 384-well plates (Corning). Cell plates were centrifuged for 1 min at 1000 rpm, then incubated at 37°C overnight. The following day cells were treated as described with various concentrations of Thapsigargin or Tunicamycin, incubated for a further 18 hr at 37°C, equilibrated to room temperature, then 20 µL of SteadyLite or Renilla-Glo (Promega, Madison, WI) were added to each well. Luminescence activity was measured 10 min after reagent addition with an EnVision Multilabel Reader (PerkinElmer) using a 100 ms integration time.

### 1536-well, high-throughput screening

HEK293T-Rex cells incorporating either the ERSE-FLuc or XBP1s-RLuc reporters were collected by trypsinization and resuspended at a density of 500,000 cells per mL. The assay was started by dispensing 5 µL of cell suspension into each well of white, solid-bottom 1536-well plates using a flying reagent dispenser (FRD) and placed into an online incubator for 3 hr. Cells were then treated with 34 nL/well of either test compounds to give final concentrations of 6.8 µM, DMSO (low control, final concentration 0.68%, 0% activation) or 37 µM of Delta-7 thapsigargin (high control, final concentration 500 nM, 100% activation). Plates were incubated for 18 hr at 37°C, removed from the incubator

and equilibrated to room temperature for 10 min. Luciferase activity was detected by addition of 5 µL of ONE-Glo reagent (Promega) to each well. After a 10 min incubation time, light emission was measured with the ViewLux reader (PerkinElmer). The percent activation of each test compound was calculated as follows: % Activation = 100*(Test Compound- Median Low Control) / (Median High Control – Median Low Control).

## Multiplex gene expression (MGE) profiling

96-well plates were seeded with HEK293T cells (20,000 cells/well) and incubated at 37°C overnight. Cells were treated with compounds in media to give a final concentration of 10 µM and incubated for 6 hr. Media was removed, then cell pellets in the plates were frozen at -80°C. On the day of assay, cell pellets were thawed, lysed and analyzed for gene expression as previously described (*Calamini et al., 2012*). RNA expression changes were $\log_2$ transformed and clustering was performed in R using a Euclidean distance function and Ward's method for hierarchical clustering.

## mRNA-Seq

HEK293T-Rex and HEK293[DAX] cells in 12-well plates were treated for 6 hr with vehicle, 1 µM Tg, 10 µM TMP (in HEK293[DAX]), or 10 µM **132**, **147** or **263** in biological triplicate at 37°C. Cells were harvested, and RNA was extracted using the RNeasy Mini Kit (Qiagen, Hilden, Germany). Total RNA was quantified using NanoDrop (ND-1000). Samples were run on the Illumina HiSeq system.

Single end, 100 bp-long reads from RNA-Seq experiments were aligned to the GRCh37.p13 human genome reference assembly using SeqMan NGen 11.2.1 (DNAStar, Inc., Madison, WI). The assembly data were then imported into ArrayStar with QSeq (DNAStar, Inc.) to quantify the gene expression levels. The sequence counts were normalized to reads per kilobase per million (RPKM) after filtering out non-mRNA sequence features. The statistical significance of the difference between the expression levels of a gene under different conditions was assessed using a Student's t-test with the Benjamini-Hochberg multiple testing correction.

## Geneset analysis of transcriptional data

ATF6- and XBP1s-selective target genes were identified from transcriptional profiles of HEK293[DAX] cells following stress-independent activation of TMP-dependent DHFR-ATF6 and/or doxycycline inducible XBP1s, as previously described (*Shoulders et al., 2013*). PERK-selective target genes were identified from *Lu et al., 2004*. These genesets are described in *Figure 3—source data 1*. Only ATF6-, XBP1s- or PERK-selective genes induced >1.5 fold in Tg-treated samples were used in this analysis. The log transformed fold-increase of these target genes in HEK293T-Rex cells treated with the respective ER proteostasis regulators was then normalized to the log transformed fold increase of these genes induced by Tg treatment. The plots shown in *Figure 3B–E* were prepared as box and whisker plots using Kaleidograph. Differential activation of the ATF6, XBP1s, and PERK genesets was assessed by one-way ANOVA and significance of pairwise comparison confirmed by unpaired t-test. An analogous strategy was employed to prepare the heat map shown in *Figure 2—figure supplement 1C* using transcriptional data from the MGE analysis.

## Proteomics by TMT-MuDPIT

HEK293T-Rex cells in 6-well plates were treated for 16 hr with vehicle, or **132**, **147** or **263** at 37°C. Lysates were prepared in radioimmunoprecipitation assay (RIPA) buffer (150 mM NaCl, 50 mM Tris pH 7.5, 1% Triton X-100, 0.5% sodium deoxycholate, and 0.1% SDS) with fresh protease inhibitor cocktail (Roche, Indianapolis, IN) and centrifuged for 20 min at 10000 x *g*. Protein concentrations of supernatants were determined by BCA (Thermo Fisher). For each sample, 100 µg of lysate was washed by chloroform/methanol precipitation. Air-dried pellets were resuspended in 1% RapiGest SF (Waters) and brought up in 100 mM HEPES (pH 8.0). Proteins were reduced with 5 mM Tris(2-carboxyethyl)phosphine hydrochloride (Thermo Fisher) for 30 min and alkylated with 10 mM iodoacetamide (Sigma Aldrich, St. Louis, MO) for 30 min at ambient temperature and protected from light. Proteins were digested for 18 hr at 37°C with 2 µg trypsin (Promega). After digestion, 20 µg of peptides from each sample were reacted for 1 hr with the appropriate TMT-NHS isobaric reagent (Thermo Fisher) in 40% (v/v) anhydrous acetonitrile and quenched with 0.4% $NH_4HCO_3$ for 1 hr. Samples with different TMT labels were pooled and acidified with 5% formic acid. Acetonitrile was

evaporated on a SpeedVac and debris was removed by centrifugation for 30 min at 18,000 x g. MuDPIT microcolumns were prepared as described (*Ryno et al., 2014*). LCMS/MS analysis was performed using a Q Exactive mass spectrometer equipped with an EASY nLC 1000 (Thermo Fisher). MuDPIT experiments were performed by 5 min sequential injections of 0, 10, 20, 30, . . ., 100% buffer C (500 mM ammonium acetate in buffer A) and a final step of 90% buffer C / 10% buffer B (20% water, 80% acetonitrile, 0.1% fomic acid, v/v/v) and each step followed by a gradient from buffer A (95% water, 5% acetonitrile, 0.1% formic acid) to buffer B. Electrospray was performed directly from the analytical column by applying a voltage of 2.5 kV with an inlet capillary temperature of 275°C. Data-dependent acquisition of MS/MS spectra was performed with the following settings: eluted peptides were scanned from 400 to 1800 m/z with a resolution of 30000 and the mass spectrometer in a data dependent acquisition mode. The top ten peaks for each full scan were fragmented by HCD using a normalized collision energy of 30%, a 100 ms activation time, a resolution of 7500, and scanned from 100 to 1800 m/z. Dynamic exclusion parameters were 1 repeat count, 30 ms repeat duration, 500 exclusion list size, 120 s exclusion duration, and exclusion width between 0.51 and 1.51. Peptide identification and protein quantification was performed using the Integrated Proteomics Pipeline Suite (IP2, Integrated Proteomics Applications, Inc., San Diego, CA) as described previously (*Ryno et al., 2014*). The geneset analysis (*Figure 3F–H*) was performed analogous to the analysis of the RNA-seq data using the same proteins from the genesets for ATF6, XBP1s and PERK and log transformed fold changes of protein expression. The plots shown in *Figure 3F–H* were prepared as box and whisker plots using Kaleidograph. Differential activation of the ATF6, XBP1s, and PERK target protein sets was assessed by one-way ANOVA and significance of pairwise comparison confirmed by unpaired t-test. The comparison of RNA-seq and proteomics data (*Figure 3—figure supplement 2A–C*) and the data for the geneset analysis (*Figure 3F–H*) are included as Source Data.

## Statistical methods

Unless otherwise noted, all p-values were calculated by performing a paired or unpaired (noted) t-test.

## Cytotoxicity assays

HEK293T-Rex cells incorporating either the ERSE-FLuc or XBP1s-RLuc reporters were collected by trypsinization and resuspended at a density of 500,000 cells per mL. The assay was started by dispensing 5 µL of cell suspension into each well of white, solid-bottom 1536-well plates using a flying reagent dispenser (FRD) and placed into an online incubator for 3 hr. Cells were then treated for 48 hr with various concentrations of proteostasis regulator, DMSO (0% toxicity) or 340 µM Doxorubicin as a positive control (100% toxicity). After 48 hr of incubation, toxicity was determined using an ATP detection method (CellTiter-Glo, Promega).

HepG2 cells were plated at 5000 cells/well in a translucent, flat-bottomed 96 well plate, and treated for 48 hr with vehicle, 10 µM Tg or 10 µM ER proteostasis regulator. ALMC-2 or KAS-6/1 cells were plated at 33,000 cells/well in 96-well plates, then treated with proteostasis regulators for 16 hr as described. Cell metabolic activity was measured using the CellTiter-Glo assay (Promega). CellTiter-Glo reagent was added to cell culture media at a 1:1 ratio and incubated for 2 min on an orbital shaker to induce cell lysis. The plate was then incubated at room temperature for 10 min to stabilize the luminescent signal and read on a Tecan F200 Pro microplate reader.

## Quantitative RT-PCR

Cells were treated as described at 37°C, washed with Dulbecco's phosphate-buffered saline, and then RNA was extracted using the RNeasy Mini Kit (Qiagen). qPCR reactions were performed on cDNA prepared from 500 ng of total cellular RNA using the QuantiTect Reverse Transcription Kit (Qiagen). The FastStart Universal SYBR Green Master Mix (Roche), cDNA, and appropriate human primers (*Shoulders et al., 2013*) or mouse primers (BiP; gtccaggctggtgtcctctc and gattatcggaagccgtggag) purchased from Integrated DNA Technologies were used for amplifications (45 cycles of 1 min at 95°C, 10 s at 95°C, 30 sec at 60°C) in an ABI 7900HT Fast Real Time PCR machine. Primer integrity was assessed by a thermal melt to confirm homogeneity and the absence of primer dimers. Transcripts were normalized to the housekeeping gene *RPLP2* and all measurements were

performed in triplicate. Data were analyzed using the RQ Manager and DataAssist 2.0 softwares (ABI, Foster City, CA).

XBP1 splicing was assessed by RT-PCR followed by gel electrophoresis on a 3% agarose gel using the following primers flanking the XBP1 splicing sites: CCTTGTAGTTGAGAACCAGG, GAGTCAA TACCGCCAGAATC.

## ATF6 nuclear localization

300 µL of 1.375 x $10^4$ U2OS-GFP-ATF6 cells per mL were plated per well in 96 well imaging plate (ibidi 89626, Madison, WI) and sealed with breathable seals (E&K Scientific T896100, Santa Clara, CA) two days prior to drug addition. Immediately prior to addition to cells, compounds were diluted to 6x in media from 500x DMSO stock and 60 µL 6x was added to cells for 1x final (0.2% DMSO).

After 5 hr, media was removed and cells were fixed in 4% PFA (Electron Microscopy Sciences 15714, Hatfield, PA) in PHEM buffer (60 mM PIPES, 25 mM HEPES, 10 mM EGTA, 2 mM MgCl$_2$-hexahydrate, pH 6.9) for 15 min at RT. Cells were permeabilized with PHEM-Tx (PHEM containing 0.1% Triton X-100, two washes, 5 min at RT), washed twice in PHEM and blocked in PHEM containing 2% normal goat serum (Jackson Immunoresearch Laboratories 005-000-121, West Grove, PA) for 1 hr at RT. Primary antibodies were incubated in blocking solution overnight at 4 °C. Cells were washed three times in PHEM-Tx then incubated with secondary antibodies and nuclear stain (DAPI, Thermo Fisher D-1306, 5 µg/ml) in blocking solution for 2 hr at RT protected from light. Cells were washed three times in PHEM-Tx, then twice in PHEM. Antibodies used were rat anti-GRP94 9G10 (Abcam ab2791, Cambridge, MA), mouse anti-GFP 3E6 (Life Technologies A11120), anti-rat-Alexa-555 (Life Technologies A21434), and anti-mouse-Alexa-488 (Life Technologies A11029), each at 1:1000 dilution.

Plate was imaged on a spinning disk confocal with Yokogawa CSUX A1 scan head, Andor iXon EMCCD camera and 20x Plan Apo Objective NA 0.79. Using the µManager high-content screening plugin 'HCS Site Generator' (*Edelstein et al., 2014*) 49 fields per well were acquired for a mean cell number per well of 368 ± 12.

Images were analyzed using CellProfiler (*Carpenter et al., 2006*), MATLAB R2014a and Graph-Pad Prism 5. Masks for the ER and nucleus of each cell were created using the GRP94 and DAPI staining, respectively. The ratio of the GFP intensity in the nucleus versus the ER was calculated for each cell and plotted as a histogram per well. A threshold for the minimum ratio of nuclear to ER signal corresponding to an activated (i.e., nuclear localized ATF6) cell was calculated as the minimum nuclear: ER ratio greater than 1 where the number of ER stressed cells (Tg) was greater than the corresponding unstressed control. Percent activation per well was calculated as the percentage of cells per well with a nuclear: ER ratio greater than the calculated threshold for that plate. Mean percent activation per well for a minimum of three replicate wells per treatment was plotted and error bars are standard error of the mean. Compounds are annotated as hits if they show percent activation more than three standard deviations away from the mean of vehicle treated control.

## [$^{35}$S] Metabolic labeling experiments

HepG2$^{ATF6}$ cells plated on poly-D-lysine coated dishes were metabolically labeled in DMEM-Cys/-Met (Corning CellGro, Mediatech Inc., Manassas, VA) supplemented with glutamine, penicillin/strep-tomycin, dialyzed fetal bovine serum, and EasyTag EXPRESS [$^{35}$S] Protein Labeling Mix (Perkin Elmer) for 30 min. Cells were washed twice with complete media and incubated in pre-warmed DMEM for the indicated times. Media or lysates were harvested at the indicated times. Lysates were prepared in RIPA buffer with fresh protease inhibitor cocktail (Roche). FLAG-tagged TTR variants were immu-nopurified using M1 anti-FLAG agarose beads (Sigma Aldrich) and washed three times with RIPA buffer. The immunoisolates were then eluted by boiling in Laemmli buffer and separated on SDS-PAGE. For total secreted media, 15 µ L aliquots from media incubated for 4 hrs on pulsed cells were run on a SDS-PAGE. The gels were then dried, exposed to phosphorimager plates (GE Healthcare, Pittsburgh, PA), and imaged with a Typhoon imager. Band intensities were quantified by densitometry in ImageQuant. Fraction secreted was calculated using the equation: fraction secreted = [extracellular [$^{35}$S]-TTR signal at t / (extracellular [$^{35}$S]-TTR signal at t=0 + intracellular [$^{35}$S]-TTR signal at t=0)]. Fraction remaining was calculated using the equation: [(extracellular [$^{35}$S]-TTR signal at t

+ intracellular [$^{35}$S]-TTR signal at t) / (extracellular [$^{35}$S]-TTR signal at t=0 + intracellular [$^{35}$S]-TTR signal at t=0)].

## Immunoblotting, SDS-PAGE and immunoprecipitation

For immunoblotting, cells were lysed in 50 mM Tris buffer, pH 7.5 containing 0.1% TritonX (Fisher Scientific) and supplemented with protease inhibitor cocktail (Roche). For immunoblots of endogenous ATF6, cells were washed in PBS containing protease inhibitor cocktail and 10 µM MG-132 and subsequently lysed in 1x Laemmli buffer, supplemented in the same way, by repeating cycles of boiling and vortexing. Protein lysate concentrations were normalized by Bradford (Bio-Rad, Hercules, CA) or BCA assay (Thermo Fisher). Lysates or media were boiled for 5 min in Laemmli buffer + 100 mM DTT before loading onto SDS-PAGE gel. Proteins were transferred from gel slabs to nitrocellulose, and the Odyssey Infrared Imaging System (Li-Cor Biosciences) was used to detect proteins of interest.

For immunoprecipitations, cells were washed with PBS then cross-linked with 0.5 mM Dithiobis (succinimidyl propionate) (DSP) for 30 min at room temperature. The reaction was quenched by addition of 100 mM Tris pH 7.5, then RIPA buffer was added to the cell pellets for lysis. Lysates were cleared by centrifugation at centrifuged at 10000 x $g$ for 15 min. Proteins were immunopurified using sheep polyclonal free λ LC antibody (Bethyl Laboratories A80-127A, Montgomery, TX) that was covalently conjugated to CNBr-activated Sepharose 4B (GE Healthcare). After four washes in RIPA buffer, proteins were eluted by boiling in Laemmli buffer + 100 mM DTT and samples were separated by SDS-PAGE and transferred to nitrocellulose membranes. Blots were probed with the following primary antibodies: monoclonal mouse M2 anti-FLAG (1:500, Sigma Aldrich), polyclonal rabbit anti-TTR (1:1000, Dako, Carpinteria, CA), rabbit polyclonal anti-human lambda light chain (1:1000, Bethyl Laboratories A90-112A), mouse monoclonal anti-Grp78 (1:500, Santa Cruz Biotechnology sc-166490, Dallas, TX), rabbit polyclonal anti-Grp94 (1:1000, GeneTex GTX103203, Irvine, CA), rabbit polyclonal anti-PDIA4 (1:1000, Proteintech Group 14712-1-AP, Rosemont, IL), mouse monoclonal anti-ATF6α (1–7) (BioAcademia, 1:1000, Osaka, Japan), rabbit anti-phospho eIF2α (CellSignaling Technologies #9721, 1:500, Danvers, MA), mouse anti-eIF2α (Abcam ab5369, 1:1000) and mouse monoclonal anti β-actin (1:10000, Sigma Aldrich).

## Light chain ELISA

ALMC-2 or KAS-6/1 plasma cells were plated 100,000 cells/well in 150 µL of media in 96-well Multi-Screen$_{HTS}$ filtration plates (EMD Millipore MSBVS1210, Billerica, MA). Triplicate wells were treated with DMSO or compounds at the indicated concentrations and incubated for 16 hr. Media was removed by filtration using a QIAvac 96 vaccuum manifold (Qiagen) and wells were washed two times with 150 µL media. Wells were then incubated with 150 µL of fresh media for 2 hr and the conditioned media was harvested into a 96-well plate using the vacuum manifold. Free LC and IgG concentrations were determined by ELISA in 96-well plates (Immulon 4HBX, Thermo Fisher). Wells were coated overnight at 37 °C with sheep polyclonal free λ LC antibody (Bethyl Laboratories, A80-127A) at a 1:500 dilution or human IgG-heavy and light chain antibody (Bethyl Laboratories, A80-118A) at a 1:2000 dilution in 50 mM sodium carbonate (pH 9.6). In between all incubation steps, the plates were rinsed extensively with Tris-buffered saline containing 0.05% Tween-20 (TBST). Plates were blocked with 5% non-fat dry milk in TBST for 1 hr at 37°C. Media analytes were diluted between 5 – 200 fold in 5% non-fat dry milk in TBST and 100 µL of each sample was added to individual wells. Light chain or IgG standards ranging from 3 – 300 ng/mL were prepared from purified human Bence Jones λ light chain or human reference serum (Bethyl Laboratories, P80-127 and RS10-110). Plates were incubated at 37 °C for 1.5 hr while shaking. Finally, HRP-conjugated goat anti-human λ light chain antibody (Bethyl Laboratories, A80-116P) was added at a 1:5,000 dilution or HRP-conjugate IgG-Fc fragment cross-adsorbed antibody (Bethyl Laboratories, A80-304P, 1:30,000 dilution) was added in 5% non-fat dry milk in TBST, followed by a 1.5 hr incubation of the plates at 37 °C. The detection was carried out with 2,2'-azinobis(3-ethylbenzothiazoline-6-sulfonic acid) (ABTS, 0.18 mg/mL) and 0.03% hydrogen peroxide in 100 mM sodium citrate pH 4.0. Detection solution (100 µL) was added to each well and the plates were incubated at room temperature. The absorbance was recorded at 405 nm and the values for the LC standards were fitted to a 4-parameter logistic

function. Light chain or IgG concentrations were averaged from at least 3 independent replicates under each treatment and then normalized to vehicle conditions.

## Conditioned media aggregation and blue-native PAGE

ALMC-2 plasma cells were plated $2 \times 10^6$ cells/well in 1 mL of media in 12-well plates. Wells were treated with DMSO or 10 µL of compounds and incubated for 16 hr. Cells were transferred to micro-centrifuge tubes and washed two times in media. Small aliquots (10 µL) of cells were removed for a cytotoxicity assay as described above. Cells were resuspended in media (1 mL for DMSO treatment) and the volume was adjusted for each treatment based on cell viability. Cells were conditioned in the media for 8 hr and then were removed by centrifugation. The media samples were immunoprecipitated overnight at 4 °C with recombinant Protein A – Sepharose 4B resin (Life Technologies) to remove fully assembled IgGs that interfere with LC aggregate detection. Cleared media samples were then heated to 55 °C for 8 hr to induce LC aggregation, added to Blue-Native PAGE loading dye (10% glycerol, 0.5% Coomassie G-250) and then loaded onto 3–12% Bis-Tris gradient gels (Life Technologies). The cathode buffer contained 50 mM Tricene and 15 mM Bis-Tris, pH 7.0 with 0.02% Coomassie G-250. The anode buffer contained 50 mM Bis-Tris pH 7.0. The gels were transferred onto PVDF membranes and LC was detected by rabbit anti-human λ light chain polyclonal antibody (Bethyl Laboratories), followed by HRP-conjugated secondary antibodies. The blots were imaged using a chemiluminescence substrate (Luminata Forte Western Luminescence Substrate, EMD Millipore) and imaged with a ChemiDoc XRS+ scanner (Bio-Rad).

## Acknowledgements

We thank Randal Kaufman for providing ATF6$^{+/+}$ and ATF6$^{-/-}$ MEFs. We thank Diane Jelinek for providing ALMC-2 and KAS-6/1 plasma cells. We also thank the NIH (AG046495, DK102635, DK046335, DK107604, NS092829), Arlene and Arnold Goldstein, the Amyloidosis Foundation, the Skaggs Institute for Chemical Biology for financial support. LP is supported by the Leukemia and Lymphoma Society. CBC is supported by the NIH (F32 AG042259).

## Additional information

### Competing interests

JWK: Reviewing editor, *eLife*, and a founder of Proteostasis Therapeutics Inc. (who is independently pursuing ATF6 activators, however Dr. Kelly is unaware of the structures of their activators, which were discovered by a totally different screening approach) and is a member of its scientific advisory board. WD, DG: Employees of Proteostasis Therapeutics Inc. The other authors declare that no competing interests exist.

### Funding

| Funder | Grant reference number | Author |
| --- | --- | --- |
| National Institutes of Health | AG046495 | R Luke Wiseman<br>Jeffery W Kelly |
| Amyloidosis Foundation | Jr. Research Grant Award | R Luke Wiseman |
| Skaggs Institute of Chemical Biology | | Jeffery W Kelly |
| Leukemia and Lymphoma Society | | Lars Plate |
| National Institutes of Health | DK102635 | R Luke Wiseman |
| National Institutes of Health | DK046335 | Jeffery W Kelly |
| National Institutes of Health | NS092829 | R Luke Wiseman |
| National Institutes of Health | AG042259 | Christina B Cooley |
| National Institutes of Health | DK107604 | R Luke Wiseman |

The funders had no role in study design, data collection and interpretation, or the decision to submit the work for publication.

## Author contributions

LP, CBC, JJC, CMG, RLW, Conception and design, Acquisition of data, Analysis and interpretation of data, Drafting or revising the article; RJP, Acquisition of data, Analysis and interpretation of data, Drafting or revising the article; FM, WD, Acquisition of data, Analysis and interpretation of data; JCG, DG, TPS, LS, ETP, PW, PH, Conception and design, Analysis and interpretation of data; SJB, HR, Conception and design, Contributed unpublished essential data or reagents; JWK, Conception and design, Analysis and interpretation of data, Drafting or revising the article

## Author ORCIDs

Lars Plate, http://orcid.org/0000-0003-4363-6116
Christina B Cooley, http://orcid.org/0000-0001-9699-7682
Peter Walter, http://orcid.org/0000-0002-6849-708X
R Luke Wiseman, http://orcid.org/0000-0001-9287-6840

# Additional files

## Supplementary files

• Supplementary file 1. Excel spreadsheet describing the parameters defining the High Throughput primary screen to identify small molecule ER proteostasis regulators.

• Supplementary file 2. Excel spreadsheet describing the toxicity of our top 8 small molecule ER proteostasis regulators in HEK293T-Rex cells.

• Supplementary file 3. Excel spreadsheet describing the structure, source, and purity for the compounds used in this manuscript.

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
