## [Decision Letter]

Thank you for submitting your article "Small Molecule Proteostasis Regulators that Reprogram the ER to Reduce Extracellular Protein Aggregation" for consideration by *eLife*. Your article has been favorably evaluated by Randy Schekman as the Senior editor and three reviewers, one of whom, Davis Ng, is a member of our Board of Reviewing Editors.

The reviewers have discussed the reviews with one another and the Reviewing Editor has drafted this decision to help you prepare a revised submission.

Summary:

The authors set out to discover small molecule modifiers of the mammalian unfolded protein response, with the ultimate goal of testing such compounds as therapeutics for some types of conformational disease. In this study, they wished to "drug" the ATF6 pathway of the UPR by specifically elevating it to enhance quality control mechanisms and prevent the secretion of amyloidogenic proteins. The approach is to screen compounds in cultured cells carrying an ATF6 specific reporter and filtering upregulation hits through secondary screens for quiescence of the other branches. In this way, no specific factor was targeted. Because the transcriptional programs of each pathway were already intensely studied, the authors use gene sets defined by previous studies to further categorize each compound. The authors screened over 600,000 compounds and found 281 that activated the ERSE reporter, with 200 of those displaying reduced activity for the IRE1 pathway. UPR target gene transcriptional profiling filtered out 79 compounds that more strongly activate the ATF6 pathway. Of those, 3 (**132**, **147**, **263**) were used for detailed analyses. Genome wide mRNAseq and proteomic profiling was performed on cells treated with these compounds. **132** is the strongest inducer but it activates all three pathways. **147** and **263** are more specific for the ATF6 pathway but does so more weakly than **132**. All three activate ATF6 processing and can reduce the secretion of TTR and ALLC amyloidogenic molecules. Co-immunoprecipitation and cyclohexemide chase experiments indicate that retention is not accompanied by enhanced ER-associated degradation.

In short, this is a carefully performed study that yielded a new class of small molecule modifiers that preferentially activate the ATF6 pathway of the mammalian UPR. For a complex screen, the experimental design is efficient and included detailed validation of the compounds' actions in cells. Although the identity of their direct targets remain unclear, these are important advances that will be of interest to the broad readership of *eLife*.

Essential revisions:

This manuscript contains a vast amount of data, which is somewhat overwhelming in places. I see the only weakness of the manuscript in the presentation of the data, but a redesign of the manuscript should be sufficient to make this work accessible for a wider audience. Specifically, it is necessary to explain the overall screen design in a clearer fashion, and explain the selection of the "preferential activators", especially since some of them do not match the criteria of the counterscreen, and don't show selectivity for ATF6 (e.g. compound 238). The third screening step (transcriptional profiling) needs to be explained in more detail. For the remainder of the manuscript it would be helpful if the authors then focus on just the compounds that are characterized in detail in the last parts of the manuscript (e.g. just **147** and perhaps additionally **263**).

1) One concern is that each of the prioritized compounds have known chemical liabilities (Table 1). The phenolic groups of catechols are often associated with promiscuous protein interactions and they have known issues related to rapid metabolism. Similarly, the nitroso- groups (**263**, **148**) are prone to reduction and the N-N and N-O bonds (5, 145, 148, 258) have been recognized as problematic at multiple stages of probe discovery and a couple compounds (**238**, **132**) have predicted unsaturated ketone-based Michael acceptors. While a full structure-activity campaign for each chemical series is well beyond the scope of the study, it seems prudent to understand whether the liabilities are essential for activity. Can the phenolic groups be protected as methyl esters? Replaced? Is activity reversible after washout or are the Michael acceptors indicative of a covalent mechanism? This issue seems important because other groups might start using these compounds to preferentially activate the ATF6 arm in their own work, so one extra step of focusing on the most promising 1-2 chemical series seems appropriate. This is especially true because the ATF6 target has already been validated by the author's previous work, so the biology per se is not breaking new ground, creating an opportunity for more advanced medicinal/chemical biology as the focus of the work. Based on the number of chemical liabilities, it seems possible that 50% or more of the prioritized series will be pruned by the immediate next step of triage.

2) The ALMC-2 data (Figure 6) shows some statistically significant and potentially exciting, effects on the secretion of light chain. To provide more confidence in the significance of the results and to understand the therapeutic window, it seems important to include dose dependence for both LC secretion, IgG levels and cell viability (Figure 6). Single concentration / single timepoint comparisons are difficult to interpret or rank order.

---

## [Author Response]

Essential revisions:

This manuscript contains a vast amount of data, which is somewhat overwhelming in places. I see the only weakness of the manuscript in the presentation of the data, but a redesign of the manuscript should be sufficient to make this work accessible for a wider audience. Specifically, it is necessary to explain the overall screen design in a clearer fashion, and explain the selection of the "preferential activators", especially since some of them do not match the criteria of the counterscreen, and don't show selectivity for ATF6 (e.g. compound **238**). The third screening step (transcriptional profiling) needs to be explained in more detail. For the remainder of the manuscript it would be helpful if the authors then focus on just the compounds that are characterized in detail in the last parts of the manuscript (e.g. just **147** and perhaps additionally **263**).

We thank the reviewers for this valuable suggestion. To address this recommendation, we have substantially rewritten the Results section describing how we prioritized our top 8 small molecule ER proteostasis regulators. Notably, our ranking of compounds was based primarily on the MGE data, which provides a more thorough and rich dataset as compared to our reporter data.

Importantly, **238** was included in the MGE counterscreen because it was identified as one of the top 281 compounds that activated ATF6 in our primary and validation screening using the ERSE-FLuc reporter. While it did not show preferential activation of the XBP1-RLuc reporter, as indicated by the reviewer, it did show preferential activation of ATF6 target genes in our MGE data. This preferential activation of ATF6 target genes combined with the unique structure of this compound is the reason this molecule was prioritized as one of our 8 compounds selected for follow-up. The changes to the text of the revised manuscript relevant to this topic are found directly below:

“As a final filter, we performed multiplex gene expression (MGE) profiling in HEK293T-Rex cells treated with the 281 hits to directly monitor transcriptional changes and identify compounds that preferentially active the ATF6 transcriptional program (Figure 1). […] Only catechols (cluster A; Figure 1—figure supplement 1) were mostly excluded, as these are known to be associated with adverse pharmacology (Baell and Holloway, 2010).”

1) One concern is that each of the prioritized compounds have known chemical liabilities (Table 1). The phenolic groups of catechols are often associated with promiscuous protein interactions and they have known issues related to rapid metabolism. Similarly, the nitroso- groups (**263**, **148**) are prone to reduction and the N-N and N-O bonds (5, 145, 148, 258) have been recognized as problematic at multiple stages of probe discovery and a couple compounds (**238**, **132**) have predicted unsaturated ketone-based Michael acceptors. While a full structure-activity campaign for each chemical series is well beyond the scope of the study, it seems prudent to understand whether the liabilities are essential for activity. Can the phenolic groups be protected as methyl esters? Replaced? Is activity reversible after washout or are the Michael acceptors indicative of a covalent mechanism? This issue seems important because other groups might start using these compounds to preferentially activate the ATF6 arm in their own work, so one extra step of focusing on the most promising 1-2 chemical series seems appropriate. This is especially true because the ATF6 target has already been validated by the author's previous work, so the biology per se is not breaking new ground, creating an opportunity for more advanced medicinal/chemical biology as the focus of the work. Based on the number of chemical liabilities, it seems possible that 50% or more of the prioritized series will be pruned by the immediate next step of triage.

We thank the editor and reviewers for highlighting this important point. We also noted that many of our prioritized molecules contain functional groups found in pan- assay interference compounds (PAINS), including those highlighted in the above comment. The concern about these types of molecules primarily results from potential off-pathway effects that could compromise experimental analysis of their efficacy. Despite the presence of these PAIN functional groups, these compounds phenocopy genetic ATF6 activation in multiple complementary assays. In addition, RNA-seq and proteomic profiling confirms that these molecules exert very little off-pathway effects. This level of compound characterization provides a wealth of information to facilitate the efforts of researchers interested in using our molecules to ensure efficient ATF6 activation in their experimental systems. Consistent with this statement, we have distributed these molecules to >10 laboratories already, all of whom have recapitulated the preferential ATF6 activation in a variety of cell systems including multiple mammalian cell lines, stem cells, and primary cells. We address the presence of these PAIN functional groups in the revised manuscript text found directly below:

“Several of our prioritized molecules include functional groups found in pan-assayinterference compounds (PAINS) (Baell and Walters, 2014), but given that these ER proteostasis regulators showed activity in multiple complementary screening assays, they are unlikely to act through off-target pathways. Consistent with this, the molecules depend on ATF6 and its canonical activation pathway for the induction of ATF6 target genes. Furthermore, the RNA-Seq and proteomics profiling analysis for **147** and **263** confirmed that these molecules exert very little off-target transcriptome changes and proteome remodelling apart from preferential ER proteostasis network activation.”

In addition, our top two molecules **147** & **263** both contain functional groups that can be metabolically oxidized to a quinone methide, which can in turn form covalent adducts with proteins at nucleophilic residues such as Cys. In **147**, this functional group is the 2-amino-*p*-cresol. We have performed a substantial amount of medicinal chemistry to define the specific importance of this 2-amino-*p*-cresol in the **147**-dependent induction of ATF6 target genes. Removal of either the methyl group or hydroxyl substituent completely ablates compound-dependent activation of the ERSE-luciferase reporter. Furthermore, replacing the methyl group with a trifluoromethyl group – which will prevent oxidation to the quinone methide – also ablates ERSE-luciferase activation. These results demonstrate that the 2-amino-*p*-cresol of **147** is required for ATF6 activation.

Furthermore, these results support a mechanism for **147** activity involving oxidation and subsequent modification of proteins involved in regulating ATF6 activity. We have taken advantage of this covalent mechanism to identify targets of **147** involved in preferential ATF6 activation. We synthesized an active **147** analog that contains an alkyne handle. Using this compound, we can isolate **147**-protein adducts that have been subsequently identified by mass spectrometry. Initial experiments using this approach reveal that **147** covalently modifies only a few proteins, most of which localize to the ER lumen. We are currently using genetic approaches to identify the specific proteins whose modification is required for **147**-dependent ATF6 activation. In addition, we are using this alkyne modified **147** to compare the mechanism of action for other ER proteostasis regulators identified through this approach using a competition-based assay. A manuscript describing these results is currently in the latter stages of preparation.

While we appreciate the reviewer’s request to include some of this medicinal chemistry in this current manuscript, presenting only a portion of these results would probably frustrate the reader, especially since we are 1-2 months away from finishing the experiments required for this follow-up manuscript. Since our medicinal chemistry efforts are emerging as a highly cohesive story to explain the mechanism of action for one of our preferential ATF6 activators (**147**), we feel that these results should not be included in this current manuscript. Instead, this cohesive story is better suited for a fast following manuscript focused on the mechanism(s) of small molecule-dependent ATF6 activation, potentially as an *eLife* Research Advance to be paired with this current manuscript. We do discuss the potential involvement of quinone methide formation in the compound **147**-dependent ATF6 activation described in this manuscript. This new discussion is included in the revised manuscript as below:

“Both **147** and **263** contain functional groups that after metabolic oxidation to aquinone methide could modify proteins covalently. This suggests that the covalent modification of ATF6 or a protein involved in the regulation of ATF6 activity (e.g. BiP or PDIs) (Nadanaka et al., 2007; Shen et al., 2002) by our small molecules could be involved in the activation mechanism. Future medicinal chemistry efforts will focus on exploring the mechanism of action of these compounds and the potential involvement of covalent protein modification for their activity.”

2) The ALMC-2 data (Figure 6) shows some statistically significant and potentially exciting, effects on the secretion of light chain. To provide more confidence in the significance of the results and to understand the therapeutic window, it seems important to include dose dependence for both LC secretion, IgG levels and cell viability (Figure 6). Single concentration / single timepoint comparisons are difficult to interpret or rank order.

We agree, and we now include a dose-response curve for **147** treatment in ALMC-2 cells measuring both viability and ALLC secretion (Figure 6). Since **263** does not significantly influence ALLC secretion at the highest concentration (Figure 6), we did not include a dose response for this molecule. Furthermore, since we have removed discussion of other molecules in this figure, as suggested by the reviewers, we do not include dose responses for these other molecules.